# WebUOT-1M: Advancing Deep Underwater Object Tracking with A Million-Scale Benchmark

**Chunhui Zhang**[1,2,3]**, Li Liu**[2],*, **Guanjie Huang**[2]**, Hao Wen**[3]**, Xi Zhou**[3]**, Yanfeng Wang**[1,4]

[1] Cooperative Medianet Innovation Center, Shanghai Jiao Tong University, Shanghai 200240, China
[2] The Hong Kong University of Science and Technology (Guangzhou), Guangzhou 511458, China
[3] CloudWalk Technology Co., Ltd, 201203, China
[4] Shanghai AI Laboratory, Shanghai 200232, China

## Abstract

Underwater Object Tracking (UOT) is essential for identifying and tracking submerged objects in underwater videos, but existing datasets are limited in scale, diversity of target categories and scenarios covered, impeding the development of advanced tracking algorithms. To bridge this gap, we take the first step and introduce WebUOT-1M, *i.e.*, the largest public UOT benchmark to date, sourced from complex and realistic underwater environments. It comprises 1.1 million frames across 1,500 video clips filtered from 408 target categories, largely surpassing previous UOT datasets, *e.g.*, UVOT400. Through meticulous manual annotation and verification, we provide high-quality bounding boxes for underwater targets. Additionally, WebUOT-1M includes language prompts for video sequences, expanding its application areas, *e.g.*, underwater vision-language tracking. Given that most existing trackers are designed for open-air conditions and perform poorly in underwater environments due to domain gaps, we propose a novel framework that uses omni-knowledge distillation to train a student Transformer model effectively. To the best of our knowledge, this framework is the first to effectively transfer open-air domain knowledge to the UOT model through knowledge distillation, as demonstrated by results on both existing UOT datasets and the newly proposed WebUOT-1M. We have thoroughly tested WebUOT-1M with 30 deep trackers, showcasing its potential as a benchmark for future UOT research. The complete dataset, along with codes and tracking results, are publicly accessible at here.

## 1 Introduction

Underwater object tracking (UOT) refers to the task of sequentially locating a submerged instance in an underwater video, given its initial position in the first frame [36, 51, 5, 1]. As a fundamental task in computer vision, it has a wide range of applications, such as marine animal conservation [5], underwater search and rescue [1], underwater photography [41], and homeland and maritime security [25]. The underwater environment usually exhibits uneven lighting conditions, low visibility, low contrast, watercolor variations, similar distractors, camouflage, and target blurring, posing distinct challenges for UOT compared to traditional open-air tracking tasks [75, 19, 31, 61, 49].

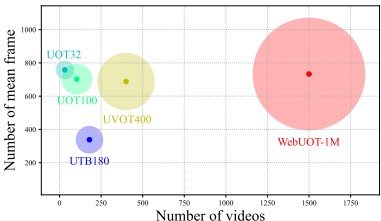

Figure 1: The proposed WebUOT-1M is much larger than existing UOT benchmarks [36, 51, 2, 1].

Despite its significance, UOT has not been thoroughly explored due to the absence of large-scale datasets, benchmarks, and challenges in gathering abundant underwater videos [1, 5].

---

*Corresponding author. E-mail: avrillliu@hkust-gz.edu.cn.

38th Conference on Neural Information Processing Systems (NeurIPS 2024) Track on Datasets and Benchmarks.

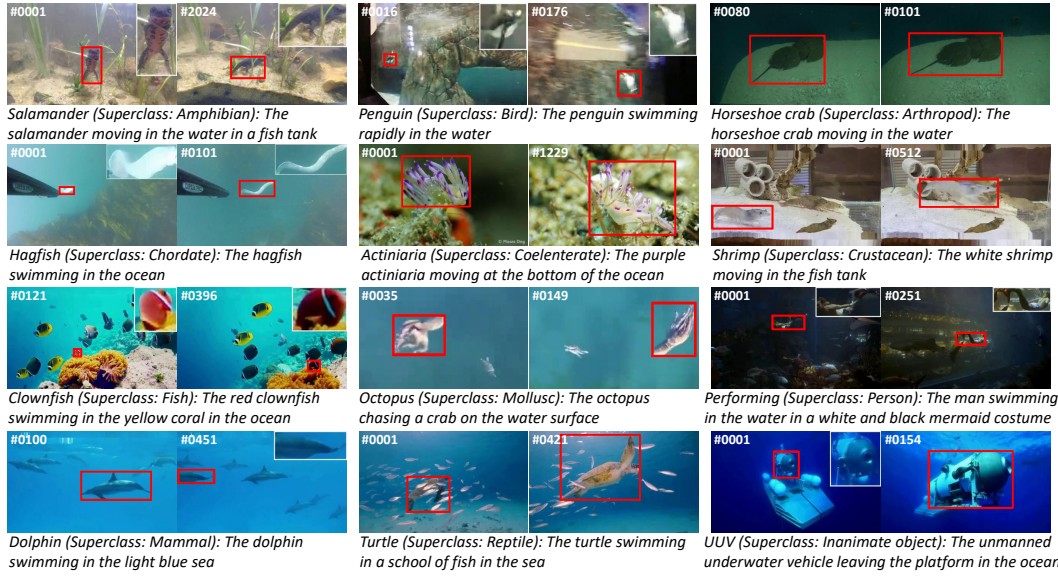

Figure 2: A glance of some video sequences and annotations from the WebUOT-1M dataset. All sequences are divided into 12 superclasses, including *amphibian, arthropod, bird, chordate, coelenterate, crustacean, fish, mollusc, person, mammal (except humans), reptile, and inanimate object*.

Recently, some efforts have been made to build UOT datasets to promote research in this field. Early works focus on specific underwater tasks (*e.g.*, tracking marine plastic waste [58]), underwater environments (*e.g.*, coral reef [20]), and specific marine species (*e.g.*, zebrafish [53] and ocean mammal [34, 37]). These datasets do help advance research on enhancing the tracking and monitoring of relevant marine species. **Due to the huge appearance variation and behavioral differences among various marine animals, models trained on these early datasets struggle with unseen species, leading to poor generalization performance.** To further facilitate research on UOT, datasets covering multiple species are proposed, *e.g.*, UTB180 [2] and UVOT400 [1]. However, these datasets are still limited in terms of their dataset volume, diversity in animal species and scenarios covered due to severe challenges in underwater video collection and annotation.

To fill this gap, we propose WebUOT-1M, the first million-scale dataset for UOT. As shown in Figs. 1 and 2, the WebUOT-1M is much larger than existing datasets and comprises abundant categories, diverse underwater scenarios, and rich annotations. WebUOT-1M comprises 1.1 million frames with precise bounding box annotations across 1,500 underwater videos and 408 highly diverse target categories (see Tab. 1, Fig. 3). These targets are further classed into 12 superclasses with reference to WordNet [47] to facilitate the evaluation of the cross-superclass generalization ability of tracking models. Most of the video clips are collected from YouTube[1] and BiliBili[2] with carefully filtering. These video websites contain massive underwater video resources. The videos are captured using different cameras, at various shooting perspectives and distances, and with different camera motion patterns. We assembled a professional labeling team to conduct data annotation. To establish a comprehensive benchmark, we offer 23 tracking attributes, *e.g.*, low resolution, fast motion, similar distractors, underwater visibility, and watercolor variations. To explore the complementary advantages of visual and linguistic modalities, we provide a language prompt for each underwater video, which can facilitate the research of multi-modal UOT. To provide a baseline method for other researchers to compare, we propose a simple yet powerful omni-knowledge distillation tracking algorithm based on knowledge distillation (KD) [29] and contrastive learning (CL) [50].

The main contribution of this work is three-fold. 1) We introduce WebUOT-1M, the first million-scale benchmark dataset featuring diverse underwater video sequences, essential for offering a dedicated platform for the development and evaluation of UOT algorithms. 2) We propose a simple yet strong **O**mni-**K**nowledge distillion **Track**ing approach, termed OKTrack, for UOT. It is the first work to explore knowledge transfer from a teacher Transformer using underwater and enhanced frames to a student Transformer in the UOT area. 3) We comprehensively benchmark the proposed approach,

[1]https://www.youtube.com/    [2]https://www.bilibili.com/

Table 1: Comparison of WebUOT-1M with popular UOT benchmarks.

| Dataset | Year | Videos | Classes | Attributes | Min frame | Mean frame | Max frame | Total frames | Annotated boxes | Total duration | Absent label | Language prompt | Data partition | Open source |
|---|---|---|---|---|---|---|---|---|---|---|---|---|---|---|
| **UOT32** [36] | 2019 | 32 | - | - | 283 | 758 | 1,573 | 24 K | 24 K | 16 min | ✗ | ✗ | Test | Proprietary |
| **UOT100** [51] | 2022 | 104 | - | 3 | 264 | 702 | 1,764 | 74 K | 74 K | 41 min | ✗ | ✗ | Test | Fully |
| **UTB180** [2] | 2022 | 180 | - | 10 | 40 | 338 | 1,226 | 58 K | 58 K | 32 min | ✗ | ✗ | Train/Test | Fully |
| **VMAT** [5] | 2023 | 33 | 17 | 13 | 438 | 2,242 | 5,550 | 74 K | 74 K | 41 min | ✗ | ✗ | Test | Fully |
| **UVOT400** [1] | 2023 | 400 | 50 | 17 | 40 | 688 | 3,273 | 275 K | 275 K | 2.6 hours | ✗ | ✗ | Train/Test | Partially |
| **WebUOT-1M** | 2024 | 1,500 | 408 | 23 | 49 | 733 | 9,985 | 1.1 M | 1.1 M | 10.5 hours | ✓ | ✓ | Train/Test | Fully |

along with representative tracking algorithms based on CNN, CNN-Transformer, and Transformer on both the newly proposed WebUOT-1M and existing UOT datasets, obtaining some valuable insights.

## 2 Related Work

**Open-air Object Tracking.** Open-air object tracking is an active research field in computer vision, aiming to learn a class-agnostic appearance model to estimate the state of an arbitrary object in open-air videos (*e.g.*, ground [19, 49, 61], UAV [75, 30, 48, 74], and indoor scenes [66, 65]) given an initial bounding box. In the past decade, significant progress has been made in open-air tracking by embracing deep neural networks. Early deep tracking paradigms include deep discriminative correlation filters [14, 13, 24, 23, 73] and Siamese networks [3, 38, 11, 27]. They usually require carefully designed online learning strategies or complex post-processing. Recently, with the development of foundation models [55, 16, 72], many advanced tracking techniques have emerged, such as unified architectures [70, 67], autoregressive models [64, 9], prompt learning [87], and diffusion [68]. All these modern deep tracking models benefit from public large-scale datasets [42, 19, 31, 49].

**Underwater Object Tracking.** UOT [36, 51, 5, 1, 2] aims to predict the location of objects submerged in various underwater environments. Recently, it has attracted increasing attention from the research community due to the underwater vision understanding and marine animal conservation demands. Yoerger *et al.* [71] propose a Mesobot platform to track slow-moving marine animals. This platform tracks jellyfish and larvaceans by building color segmentation and blob-tracking [4] methods. However, these methods can only be used for simple underwater scenarios and a few species. Katija *et al.* [35] further propose using tracking-by-detection and deep neural networks to improve tracking robustness and adaptability in more complex underwater environments. Li *et al.* [41] introduce underwater images and open-air sequence hybrid training and motion-based post-processing to address the sample imbalance and model drift problems, respectively. To promote the research of UOT, many datasets are established, *e.g.*, UOT32 [36], UOT100 [51], UTB180 [2], VMAT [5], and UVOT400 [1]. However, these datasets either lack training sets [36, 51, 5] (see Tab. 1) or are difficult to train models with good generalization capabilities due to limited size and scenarios covered [2, 1]. To the best of our knowledge, there is still no public million-scale benchmark specifically dedicated to the UOT task. We believe that our benchmark can greatly facilitate the research of UOT.

**Knowledge Distillation.** KD [29, 26, 52, 80], *i.e.*, efficiently learning a small student network from a large teacher network, is a widely studied area. Its core idea is that the student network imitates the teacher network to obtain competitive or even better performance [26]. In recent years, there are many KD-based deep trackers, which include but do not limit to channel distillation [23], training-set distillation [39], cross-modality distillation [81, 62], correlation filter distillation [8], and Siamese network distillation [57, 84]. Inspired by recent RGB-event distillation method [62], we propose a novel omni-knowledge distillation approach in the UOT area by combining token contrastive representation, similarity matrix, feature embeddings, and response maps distillation losses.

## 3 Dataset

### 3.1 Data Collection and Annotation

The goal of WebUOT-1M is to provide a large-scale benchmark for UOT in various real-world underwater scenarios. To this end, we mainly resort to online video platforms, *e.g.*, YouTube and BiliBili, and carefully collect and filter 1,500 video sequences covering 408 different categories from abundant underwater scenarios, *e.g.*, sea, river, lake, pool, aquarium, fish tank, bowl, and cup. The video platforms contain massive real-world videos captured from different devices/platforms (*e.g.*, diver-based cameras, human-occupied vehicles, underwater robots, and mobile phones), with different shooting angles, distances, and camera movement patterns, greatly reducing the cost of collecting large-scale UOT datasets. Then, we perform data cleaning to discard videos that are not suitable for tracking, *e.g.*, repeated scenes, long-term static targets, and incomplete trajectories. The number of videos in each class group forms a long-tail distribution (see Fig. 3), which meets real-world

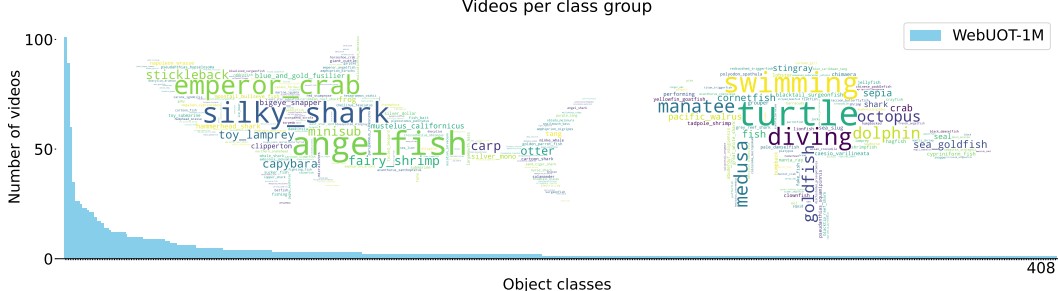

Figure 3: We propose a challenging benchmark containing diverse object classes shown in word clouds, and the number of videos in each class group forms a long-tail distribution.

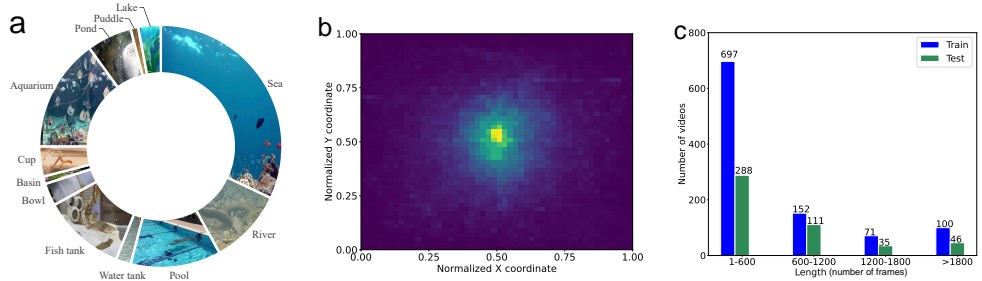

Figure 4: Statistics of WebUOT-1M. (a) Abundant underwater scenarios. (b) Distribution of normalized target center position. (c) Distribution of video length.

situations, making WebUOT-1M more challenging and encouraging the learning of more general UOT algorithms. Moreover, this brings a significant advantage to our dataset, since WebUOT contains a wide range of categories, especially many *rare species* (*e.g.*, flame squid, salamander, and Chinese sturgeon), which can facilitate the visual observations and tracking of these rare species. All videos according to the target are divided into 12 superclasses with reference to WordNet [47], *i.e.*, *amphibian, arthropod, bird, chordate, coelenterate, crustacean, fish, mollusc, person, mammal (except humans), reptile, and inanimate object* (see Fig. 2). Unlike most previous UOT datasets [36, 51, 5, 2] that only contain marine animals, WebUOT-1M incorporates inanimate objects (*e.g.*, unmanned underwater vehicle, and amphibious drone), resulting in a more comprehensive and versatile benchmark.

After the video cleaning, we randomly select moving targets in the videos to ensure the diversity of the dataset (see Fig. 2). Then, a professional data labeling team conducts multiple rounds of manual annotation and correction. *However, directly annotating some underwater videos with severe color deviation and blurring is extremely difficult or even impossible.* To address this issue, we provide the annotators with enhanced videos using a semi-supervised method [32]. The author team performs the last data verification to ensure the high quality of the annotations. Specifically, the bounding box $[x, y, w, h]$ is used as the ground-truth of the target in each frame of the video, where $(x, y)$, $w$, and $h$ represent the target's top-left corner, width, and height, respectively. Following [75, 61, 18], a sentence of language prompt describing the color, behavior, attributes, and surroundings of the target is given for each video sequence to encourage the exploration of multi-modal UOT. The *absent* label for each frame is also annotated to provide rich information for accurate tracking (see Tab. 1).

### 3.2 Attributes

To enable comprehensive and in-depth evaluation of trackers [1, 75], we label each video sequence with rich attributes. Specifically, we provide 23 attributes, including low resolution (LR), fast motion (FM), scale variations (SV), aspect ratio variations (ARV), camera motion (CM), viewpoint changes (VC), partial occlusion (PO), full occlusion (FO), out-of-view (OV), rotation (ROT), deformation (DEF), similar distractors (SD), illumination variations (IV), motion blur (MB), partial target information (PTI), natural or artificial object (NAO), camouflage (CAM), underwater visibility (UV), watercolor variations (WCV), underwater scenarios (US), shooting perspective (SP), size (SIZ), and length (LEN) of video. The detailed definition and statistics of attributes are shown in **Appendices**.

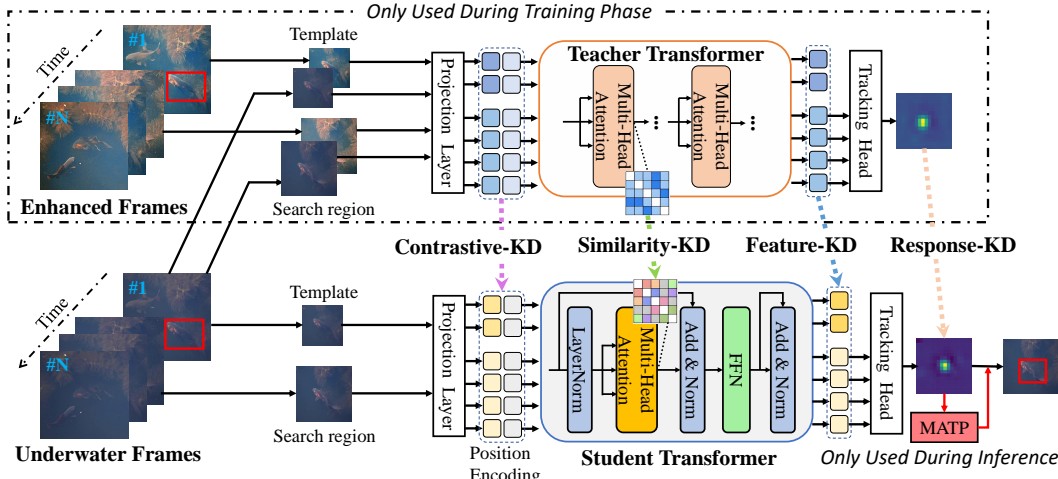

Figure 5: OKTrack overview. During training phase, we adopt four distillation losses (see Sec. 4.3). A training-free MATP module (see Sec. 4.2) is used to enhance the tracking robustness of inference.

## 3.3 Statistical Analysis

As shown in Fig. 4(a), WebUOT-1M contains abundant underwater scenarios, including sea, river, lake, pool, fish tank, water tank, basin, bowl, cup, aquarium, pond, and puddle. The normalized target center position distribution presents a center mean Gaussian (see Fig. 4(b)), indicating the high quality and diversity of the dataset. The distribution of video length is demonstrated in Fig. 4(c). We can see that WebUOT-1M contains 985, 263, 106, and 146 videos with segments containing 1-600, 600-1200, 1200-1800, and more than 1800 frames, respectively. The various video lengths make our dataset suitable for benchmarking both short-term and long-term tracking algorithms.

## 4 Methodology

In this section, we present an omni-knowledge distillation framework (see Fig. 5). The core insight is to leverage a teacher model pre-trained on the massive open-air data to enhance tracking performance on UOT. KD [29, 26] has been proven to efficiently learn a student model from a teacher model. Inspired by recent RGB-event tracking [62], we propose to distill the knowledge from enhanced and underwater frames with the supervision of a pre-trained teacher tracker to a student tracker devoted to handling underwater frames. Due to the limited space, more details are provided in **Appendices**.

### 4.1 Problem Formulation

Given underwater frames $\mathcal{I} = \{I_i\}_{i=1}^N \in \mathbb{R}^{H \times W \times C}$, we first adopt an off-the-shelf semi-supervised method [32] to obtain corresponding enhanced frames $\mathcal{E} = \{e_i\}_{i=1}^N \in \mathbb{R}^{H \times W \times C}$, where $(H, W)$ denotes the resolution of video frames, $C$ and $N$ represent the number of channels and video frames, respectively. Like Transformer-based trackers [70, 9], we can crop a pair of images patches, *i.e.*, the template patch $z \in \mathbb{R}^{H_z \times W_z \times C}$ and the search region patch $x \in \mathbb{R}^{H_x \times W_x \times C}$, from underwater frames. Then, the image patches are divided into multiple non-overlapping patches, which will be further transformed into 1D tokens using a projection layer [70]. To preserve position information, learnable positional embeddings [17] are added to these tokens. Given an underwater video with a pair of template and search region patches $\mathcal{X}_{xz} = \{x, z\}$, and an initial target box $\mathcal{B}_0$, the UOT problem can be formulated as $\mathcal{S} : \{\mathcal{X}_{xz}, \mathcal{B}_0\} \to \mathcal{B}$, where $\mathcal{S}$ is the student tracker, $\mathcal{B}$ is the predicted target box in subsequent frames. Adding enhanced template and search region patches $\tilde{\mathcal{X}}_{xz} = \{\tilde{x}, \tilde{z}\}$ and a pre-trained teacher tracker $\mathcal{T}$, the learning of the student tracker can be expressed as optimizing $\mathcal{L}_{OKD} : \{\mathcal{X}_{xz}, \tilde{\mathcal{X}}_{xz}, \mathcal{T}, \mathcal{S}_0\} \to \mathcal{S}$, where $\mathcal{L}_{OKD}$ represents the omni-knowledge distillation loss (see Sec. 4.3), $\mathcal{S}_0$ is an initial student model based on the plain ViT backnone [17].

### 4.2 Network Architecture

**Multi-view Teacher Network.** We adopt a modified version of unified backbone [79] as the teacher Transformer, which consists of multiple Transformer layers. The teacher network can use multi-view modalities simultaneously, *i.e.*, underwater and enhanced frames. Specifically, tokens from underwater and enhanced frames are concatenated and fed into the teacher Transformer. Then, the extracted feature embeddings are used for target prediction by a tracking head.

**Unimodal Student Network.** To realize efficient and low-latency UOT, the student network [70] only uses underwater video streams. As shown in Fig. 5, the student Transformer is a plain ViT architecture [17]. We argue that through the omni-knowledge distillation, an accomplished teacher can effectively transfer the knowledge obtained from handling underwater and enhanced frames to the student network, significantly enhancing the tracking performance of the student network.

**Motion-aware Target Prediction.** The underwater targets (*e.g.*, fish and sharks) are often surrounded by similar distractors, leading to model drift [41]. To tackle this issue, we design a *training-free* motion-aware target prediction (MATP) (see Fig. 5) based on Kalman filtering [33]. It involves two primary stages: *prediction*, which estimates the current state using the previous state, and *correction*, which combines the estimated state with current observations to determine the optimal state.

## 4.3 Omni-Knowledge Distillation

**Token-based Contrastive Distillation (CKD).** The CKD is employed to explicitly align underwater and enhanced tokens by CL [50], aiming to mitigate the distribution discrepancies between multi-view modalities (*i.e.*, underwater and enhanced frames). We define underwater tokens for student network as $\mathbf{t}^s \in \mathbb{R}^{320}$, underwater and enhanced tokens for teacher network as $\mathbf{t}^t \in \mathbb{R}^{320}$ and $\tilde{\mathbf{t}}^t \in \mathbb{R}^{320}$, respectively. Formally, the CKD losses between teacher and student networks are defined as:

$$\mathcal{L}_{u2e} = -\frac{1}{K}\sum_{i=1}^{K}\log\frac{\exp(sim(\mathbf{t}_i^s, \tilde{\mathbf{t}}_i^t)/\tau)}{\sum_{j=1}^{K}\exp(sim(\mathbf{t}_i^s, \tilde{\mathbf{t}}_j^t)/\tau)}], \quad \mathcal{L}_{e2u} = -\frac{1}{K}\sum_{i=1}^{K}\log\frac{\exp(sim(\tilde{\mathbf{t}}_i^t, \mathbf{t}_i^s)/\tau)}{\sum_{j=1}^{K}\exp(sim(\tilde{\mathbf{t}}_i^t, \mathbf{t}_j^s)/\tau)}]. \quad (1)$$

where $K$ is the batch size, $\tau$ is a temperature parameter, $sim(\cdot)$ denotes cosine similarity function. The CKD losses, *i.e.*, $\mathcal{L}'_{u2e}$ and $\mathcal{L}'_{e2u}$, with the teacher network can be calculated similarly. The total CKD loss is $\mathcal{L}_{CKD} = \mathcal{L}_{u2e} + \mathcal{L}_{e2u} + \mathcal{L}'_{u2e} + \mathcal{L}'_{e2u}$.

**Similarity-based Distillation (SKD).** The similarity matrix (*i.e.*, dot product of query and key) in the multi-head attention can capture rich long-range dependencies and cross-modal information [62, 17]. Given the similarity matrices (*i.e.*, $S_i^t$ and $S_i^s$) of the $i^{th}$ layer of teacher and student Transformers. We define the SKD loss as $\mathcal{L}_{SKD} = \sum_{i=1}^{L}(S_i^t - S_i^s)^2$, where $L$ is the number of Transformer layers.

**Feature-based Distillation (FKD).** The advanced feature embeddings contain rich semantic information. The FKD loss between teacher and student networks can be formalized as $\mathcal{L}_{FKD} = ||F^t - F^s||^2$, where $F^t$ and $F^s$ are feature representations of them, respectively.

**Response-based Distillation (RKD).** In general, directly mimicking the response map $R^t$ of the teacher network used for target localization enables the student network to achieve better tracking accuracy [57]. Following [62], we adopt the Gaussian weighted focal loss function $\mathcal{L}_{GWF}(\cdot)$ to define the RKD loss as $\mathcal{L}_{RKD} = \mathcal{L}_{GWF}(R^t/\mu, R^s/\mu)$, where $\mu$ is a scale factor.

Therefore, the omni-knowledge distillation loss is formulated as $\mathcal{L}_{OKD} = \mathcal{L}_{CKD} + \mathcal{L}_{SKD} + \mathcal{L}_{FKD} + \mathcal{L}_{RKD}$. We also borrow the tracking loss function used in [70, 40] (*i.e.*, GIoU loss $\mathcal{L}_{GIoU}$, focal loss $\mathcal{L}_{focal}$, and $L_1$ loss $\mathcal{L}_{L1}$) to enhance the convergence of training. Finally, the total loss can be written as $\mathcal{L}_{total} = \mathcal{L}_{OKD} + \lambda_1 \mathcal{L}_{GIoU} + \lambda_2 \mathcal{L}_{focal} + \lambda_3 \mathcal{L}_{L1}$, where $\lambda_1, \lambda_2, \lambda_3$ are balance factors.

# 5 Experiments

## 5.1 Implementation Details

We adopt the unified tracking model [79] as the teacher network. The student network [70] is based on a plain ViT-base backbone, stacked by $L$ (*i.e.*, 12) transformer encoder layers. We use an AdamW optimizer [43] with initial learning rate $4 \times 10^{-4}$. The weight decay factor is $1 \times 10^{-4}$ after 240 epochs. The batch size and total epoch are 32 and 300. The temperature parameter $\tau$ is 0.5, and the scale factor $\mu$ is empirically set to 2. Following [62], the balance factors $\lambda_1, \lambda_2, \lambda_3$ are set to 1, 1, and 14, respectively. For proper and fast verification, models are trained for 50 epochs in ablation experiments. Our experiment platform is an Ubuntu server with 8 NVIDIA A6000 GPUs.

## 5.2 Metrics and Protocols

Following [75, 18], we perform the one-pass evaluation (OPE) and measure trackers using five evaluation metrics (*i.e.*, percision (Pre), normalized precision (nPre), success rate (AUC), complete success rate (cAUC), and mean accuracy (mACC)) under two protocols.

**Protocol I.** In this protocol, we conduct a *cross-domain evaluation* of existing tracking models trained on *open-air* tracking datasets. We report the results of different trackers on the WebUOT-1M test set. Cross-domain evaluation helps ascertain how well a tracker can adapt to new and unseen data distributions, providing insights into its robustness.

**Protocol II.** In this protocol, we perform *within-domain evaluation* of open-resource trackers on the WebUOT-1M dataset. Concretely, we retrain different trackers on the training set and evaluate them on the test set. Protocol II aims to provide benchmark results for the underwater tracking community to train and evaluate trackers using a large number of underwater videos.

## 5.3 Evaluated Trackers

To provide baseline results for future research, we extensively evaluate 30 deep trackers, including *CNN-based* (*e.g.*, SiamFC [3], ECO [14], VITAL [59], ATOM [13], SiamRPN++ [38], Ocean [83]), *CNN-Transformer-based* (*e.g.*, TrDiMP [60], TransT [10], STARK-ST50 [69], ToMP-101 [45]), *Transformer-based* methods (*e.g.*, OSTrack [70], SimTrack-B32 [7], MixFormerV2-B [12], SeqTrack-B256 [9]), vision-language (VL) trackers (*e.g.*, VLT$_{TT}$ [28], JointNLT [86], CiteTracker-256 [40], All-in-One [79]), and UOT trackers (UOSTrack [41], OKTrack), as shown on Tab. 2.

Table 2: Summary of open-air and underwater tracking algorithms. "Trans." denotes Transformer. "B" represents base model.

| Tracker | Source | Backbone | FPS | UOT |
|---|---|---|---|---|
| SiamFC [3] | ECCVW16 | AlexNet | 86 | ✗ |
| ECO [14] | CVPR17 | VGG-M | 8 | ✗ |
| VITAL [59] | CVPR18 | VGG-M | 1.5 | ✗ |
| ATOM [13] | CVPR19 | ResNet-18 | 30 | ✗ |
| SiamRPN++ [38] | CVPR19 | ResNet-50 | 35 | ✗ |
| SiamBAN [11] | CVPR20 | ResNet-50 | 40 | ✗ |
| SiamCAR [27] | CVPR20 | ResNet-50 | 52 | ✗ |
| Ocean [83] | ECCV20 | ResNet-50 | 58 | ✗ |
| PrDiMP [15] | CVPR20 | ResNet-50 | 30 | ✗ |
| TrDiMP [60] | CVPR21 | ResNet-50, Trans. | 26 | ✗ |
| TransT [10] | CVPR21 | ResNet-50, Trans. | 50 | ✗ |
| STARK-ST50 [69] | ICCV21 | ResNet-50, Trans. | 40 | ✗ |
| KeepTrack [46] | ICCV21 | ResNet-50 | 18 | ✗ |
| AutoMatch [82] | ICCV21 | ResNet-50 | 50 | ✗ |
| TCTrack [6] | CVPR22 | AlexNet | 126 | ✗ |
| ToMP-101 [45] | CVPR22 | ResNet-101, Trans. | 25 | ✗ |
| AiATrack [21] | ECCV22 | ResNet-50, Trans. | 38 | ✗ |
| SimTrack-B32 [7] | ECCV22 | ViT-B | 30 | ✗ |
| OSTrack [70] | ECCV22 | ViT-B | 105 | ✗ |
| MixFormerV2-B [12] | NeurIPS23 | ViT-B | 165 | ✗ |
| GRM [22] | CVPR23 | ViT-B | 45 | ✗ |
| SeqTrack-B256 [9] | CVPR23 | ViT-B | 40 | ✗ |
| VLT$_{SCAR}$ [28] | NeurIPS22 | ResNet-50, Bert-B | 43 | ✗ |
| VLT$_{TT}$ [28] | NeurIPS22 | ResNet-50, Bert-B | 35 | ✗ |
| JointNLT [86] | CVPR23 | Swin-B, Bert-B | 39 | ✗ |
| CiteTracker-256 [40] | ICCV23 | ViT-B, CLIP | 24 | ✗ |
| All-in-One [79] | ACM MM23 | ViT-B, Bert-B | 60 | ✗ |
| UVLTrack [44] | AAAI24 | ViT-B, Bert-B | 57 | ✗ |
| UOSTrack[1] [41] | arXiv23 | ViT-B | 110 | ✓ |
| OKTrack | Ours | ViT-B | 115 | ✓ |

[1] For a fair comparison, we fine-tune this tracker on WebUOT-1M.

## 5.4 Evaluation Results

**Overall Performance.** Figs. 6 and 7 demonstrate the cross-domain evaluation results of 30 deep trackers on WebUOT-1M. We have the following observations. **1)** The top-5 trackers (*i.e.*, OKTrack, UOSTrack, All-in-One, GRM, OSTrack) are based on Transformer [17], indicating that exploring advanced architectures is still a promising direction for tracking [54]. **2)** The UOT trackers (*i.e.*, OKTrack, UOSTrack) using the plain ViT backbone surpasses state-of-the-art (SOTA) trackers [40, 9] for open-air tracking. The possible reason is that there is a huge domain gap between underwater and open-air environments. **3)** The VL tracker (*i.e.*, All-in-One) achieves the best results among open-air trackers, demonstrating that using the additional language modality can enhance tracking performance.

**Attribute-based Performance.** To comprehensively analyze the trackers facing different tracking attributes, we conduct an attribute-based evaluation using 23 attributes. The results show that the SOTA trackers still have significant room for improvement in various challenging attributes, *e.g.*, IV, SD, CAM, and PTI (see Fig. 8). More attribute-based results are shown in **Appendices**.

**Retraining Experiments.** In Tab. 3, we retrain four representative deep trackers (*i.e.*, SiamFC, ATOM, OSTrack, CiteTracker-256) on the WebUOT-1M. The results indicate that compared with the original models, the retraining models can effectively reduce the domain gap between underwater and open-air environments. This reveals the great value of the proposed WebUOT-1M dataset for developing more powerful deep UOT algorithms.

**Underwater Vision-Language Tracking.** Previous UOT datasets lack language prompt annotations [36, 51, 2, 1]. In this work, we perform a pioneering exploration of *underwater VL tracking* through carefully annotated language prompts. From Tab. 4, we make the following observations. **1)** The usage of more cues (*e.g.*, language prompt and bounding box) can significantly boost tracking performance. **2)** The language prompt-only methods [44, 86] achieve poor results on WebUOT-1M, similar to existing multi-modal open-air tracking datasets [61, 75, 78], indicating that multi-modal tracking is far from being explored. We expect that the proposed WebUOT-1M dataset can inspire the community to develop multi-modal underwater tracking algorithms.

## 5.5 Ablation Study

**Component Analysis.** The impact of four distillation strategies (*i.e.*, CKD, SKD, FKD, and RKD) is shown in Tab. 5. Each distillation strategy brings performance improvements compared to the baseline model on WebUOT-1M and UTB180. We can observe that the RKD strategy offers a greater improvement compared to the other three distillation strategies because it directly allows the student model to mimic the response map of the teacher model for target localization.

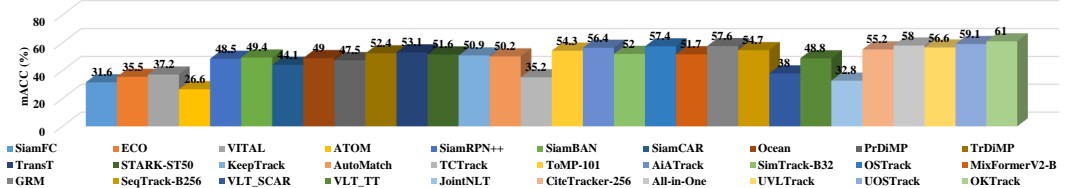

Figure 6: Overall performance on WebUOT-1M using mACC score. Best viewed by zooming in.

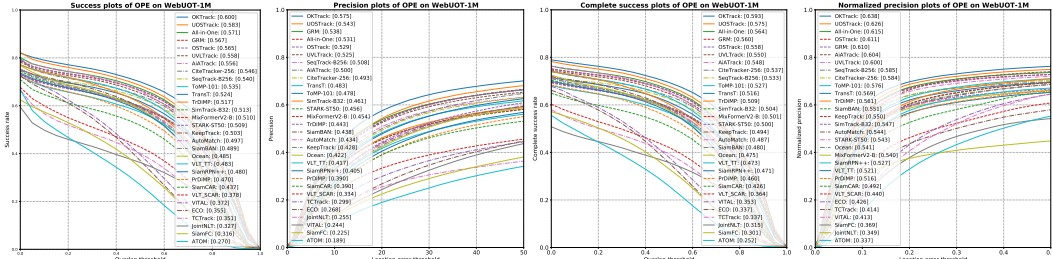

Figure 7: Overall performance on WebUOT-1M using AUC, Pre, cAUC, and nPre scores.

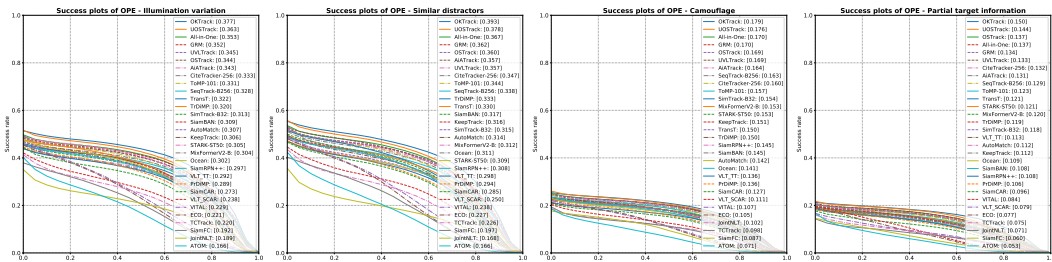

Figure 8: Evaluation results of four tracking attributes on WebUOT-1M using AUC score.

**Analysis on Motion-aware Target Prediction.** The MATP module was introduced to mitigate model drift, as underwater targets (such as fish) are often surrounded by similar distractors. We conduct experiments shown in Tab. 5. It can be observed that MATP brought gains on UTB180 and WebUOT-1M, verifying the effectiveness of MATP.

**Architecture of Student Transformer.** The main architecture of the student model is the ViT network. We conduct experiments, shown in Tab. 6, to explore the impact of different Transformer layers. We can find that increasing the number of Transformer layers (from 4 to 12) significantly improves performance, but it also increases the number of parameters, model complexity, and reduces speed. To balance performance and cost, we use a student model with 12 Transformer layers.

## 5.6 Further Discussions

**Retraining *vs.* Fine-tuning *vs.* Omni-Knowledge Distillation.** In Fig. 9, we compare three different training settings: *retraining* the student tracker using WebUOT-1M and open-air tracking datasets, *fine-tuning* the student tracker, and adopting *omni-knowledge distillation* for the student tracker on WebUOT-1M. The omni-knowledge distillation achieves the best performance. Fine-tuning the model is preferable to retraining it, as the former can mitigate the issue of insufficient data to some extent, while the latter is limited by the sample imbalance between underwater and open-air objects.

**Comparison to Other UOT Benchmarks.** We experimentally compare WebUOT-1M with three open-source UOT datasets [51, 2, 5]. From Figs. 7 and 10, we obtain some valuable insights. **1)** OKTrack achieves the best results on UTB180 and VMAT, and a comparable result on UOT100. The possible reason, as noted in [2], is that UOT100 contains a large amount of annotation errors. **2)** Compared with existing UOT datasets, WebUOT-1M is a more challenging and comprehensive benchmark suitable for both short-term tracking [51, 2] and long-term tracking [5]. **3)** The relatively poor result on the long-term tracking dataset VMAT (see Fig. 10) indicates that long-term tracking is still challenging. One solution is to utilize the rich temporal information in video sequences.

Table 3: **Retraining with WebUOT-1M.** Pre/AUC scores are reported.

| Method | UTB180 | | WebUOT-1M | |
|---|---|---|---|---|
| | Original | Retraining | Original | Retraining |
| ATOM [13] | 19.3/31.4 | 25.7/36.5 | 18.9/27.0 | 21.4/32.6 |
| SiamFC [3] | 22.3/35.1 | 27.5/39.1 | 22.5/31.6 | 25.8/38.2 |
| OSTrack [70] | 57.0/62.9 | 60.8/63.2 | 52.9/56.5 | 55.1/57.0 |
| CiteTracker-256 [40] | 54.5/61.7 | 61.6/66.3 | 49.3/54.6 | 54.2/57.7 |
| OKTrack (Ours) | -/- | 67.3/69.7 | -/- | 57.5/60.0 |

Table 4: **Vision-language tracking.** SOTA VL trackers are compared on WebUOT-1M.

| Method | Pre | nPre | AUC | cAUC | mACC |
|---|---|---|---|---|---|
| Language prompt | | | | | |
| JointNLT [86] | 22.4 | 32.2 | 31.2 | 29.8 | 31.2 |
| UVLTrack [44] | 22.5 | 33.8 | 31.2 | 30.1 | 31.3 |
| Language prompt + bounding box | | | | | |
| JointNLT [86] | 25.5 | 34.9 | 32.7 | 31.5 | 32.8 |
| VLT$_{SCAR}$ [28] | 33.4 | 44.0 | 37.8 | 36.4 | 38.0 |
| VLT$_{TT}$ [28] | 41.7 | 52.1 | 48.3 | 47.3 | 48.8 |
| CiteTracker-256 [40] | 49.3 | 58.4 | 54.6 | 53.7 | 55.2 |
| UVLTrack [44] | 52.5 | 60.0 | 55.8 | 55.0 | 56.6 |
| All-in-One [79] | 53.1 | 61.5 | 57.1 | 56.4 | 58.0 |

Table 5: **Component analysis.** Pre/AUC scores are reported on UTB180 and WebUOT-1M.

| Base | CKD | SKD | FKD | RKD | MATP | UTB180 | WebUOT-1M |
|---|---|---|---|---|---|---|---|
| ✓ | | | | | | 62.3/66.6 | 52.0/56.5 |
| ✓ | ✓ | | | | | 63.6/67.9 | 53.9/57.8 |
| ✓ | | ✓ | | | | 63.2/67.4 | 53.5/57.2 |
| ✓ | | | ✓ | | | 63.2/66.9 | 53.2/57.2 |
| ✓ | | | | | ✓ | 65.0/68.1 | 54.8/57.9 |
| ✓ | ✓ | ✓ | ✓ | ✓ | | 65.4/68.3 | 55.2/58.3 |
| ✓ | ✓ | ✓ | ✓ | ✓ | ✓ | 66.0/68.5 | 56.1/58.9 |

Table 6: **Architecture of student Transformer.** Pre/AUC scores are reported.

| #Layers | #Params | FLOPs | FPS | UTB180 | WebUOT-1M |
|---|---|---|---|---|---|
| 4 layers | 35.4 M | 10.6 G | 229 | 47.2/57.4 | 39.3/48.0 |
| 8 layers | 63.8 M | 16.8 G | 154 | 56.2/63.2 | 47.9/54.0 |
| 12 layers | 92.1 M | 21.5 G | 115 | 66.0/68.5 | 56.1/58.9 |

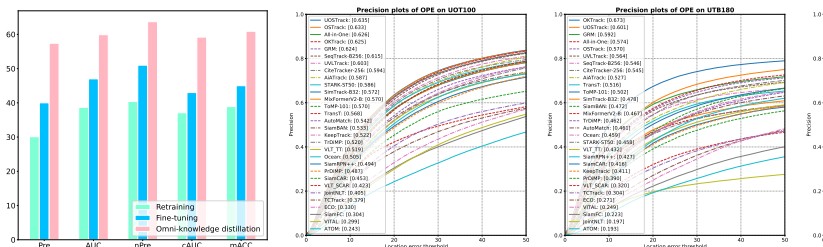

Figure 9: **Comparison of three training settings.**

Figure 10: **Evaluation on existing UOT benchmarks.** Pre scores are reported on three open-source datasets, *i.e.*, UOT100, UTB180, and VMAT.

**Stability Against Frame Rate Reduction.** In practical applications of UOT, especially in platforms of underwater unmanned robots, the need to save energy or reduce computational load often results in low frame rates [85], significantly exacerbating the challenges posed by watercolor deviation, blurring, and dynamic targets. To simulate frame rate reduction, we randomly discard some video frames and evaluate the tracking performance of different trackers on the remaining video frames. Fig.11 demonstrates the tracking performance on WebUOT-1M of five deep trackers (*i.e.*, OSTrack, CiteTracker-256, SeqTrack-B256, UOSTrack, OKTrack) with reduced frame rates, from the default frame rate (30 FPS) to the extreme thirtieth (1 FPS). We can observe that the proposed OKTrack exhibits better tracking stability in the face of video frame rate degradation.

**Tracking in Complex Underwater Scenarios.** As mentioned earlier, compared with open-air tracking, UOT presents many distinct challenges, especially watercolor variations, low underwater visibility, dense similar distractors, and camouflage that often appear simultaneously in underwater scenarios. Fig. 12 shows that the proposed OKTrack can achieve more accurate tracking in complex underwater scenarios, *e.g.*, partial target information and low underwater visibility in *shark-1*, similar distractors and occlusion in *fish-1*, compared to the other five SOTA methods [86, 40, 41, 28, 9]. This is thanks to OKTrack gaining the ability to address multi-view modalities from the teacher model, so it can achieve better performance in underwater scenarios. In addition, the MATP module makes OKTrack more robust to similar distractors, appearance changes, *etc*.

**Integrating Language Modality.** We further expand the proposed OKTrack to enable it to process both visual and language modalities, to verify that it is a flexible and scalable baseline tracker that is not only suitable for pure visual-based UOT but can also be seamlessly extended to underwater VL tracking, a new multi-modal tracking task. Specifically, we utilize the Bert-B [16] as the language encoder and employ the modal mixup [28] as the multi-modal fusion manner. The fused features are then fed into the ViT backbone for feature integration and learning. We use the weights of OKTrack, pre-trained on the visual modality, as initialization, and then fine-tune the language encoder and tracking head with both visual and language modalities to obtain the final OKTrack++. The reason we adopt a two-step training strategy ("pretraining then fine-tuning") is that our WebUOT-1M is still small compared to existing open-air tracking datasets (*e.g.*, TrackingNet [49], LaSOT [19],

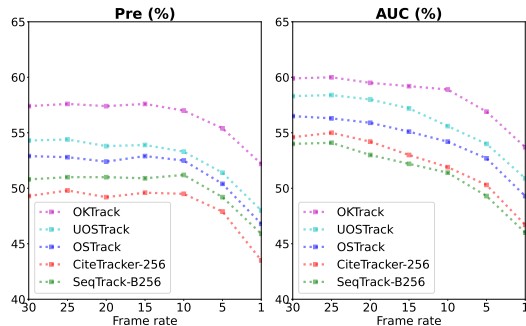

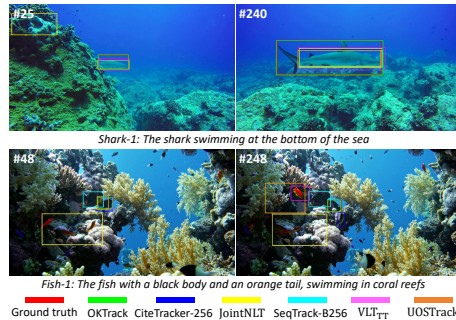

Figure 11: Stability comparison against frame rate reduction between SOTA methods on WebUOT-1M. Better stability indicates better robustness in various underwater environments.

Figure 12: Qualitative results of six representative deep trackers on WebUOT-1M. OKTrack demonstrates high tracking quality in complex underwater scenarios.

Table 7: Comparison of our two baseline trackers (OKTrack and OKTrack++) using visual and VL modalities, respectively. AUC /Pre/cAUC/nPre/mACC scores are reported on WebUOT-1M.

| Method | Type | #Params | FLOPs | FPS | WebUOT-1M |
|---|---|---|---|---|---|
| OKTrack | Visual-based | 92.1 M | 21.5 G | 115 | 60.0/57.5/59.3/63.8/61.0 |
| OKTrack++ | VL-based | 150.9 M | 57.9 G | 66 | 63.4/58.4/62.9/68.5/64.4 |

GOT-10k [31], and TNL2K [61]). Simultaneously training a visual ViT backbone (ViT-B) and a language encoder (Bert-B) is challenging to converge. Therefore, we adopted a two-step training strategy for our VL-based tracker OKTrack++. The results of our two baseline trackers (OKTrack and OKTrack++) are presented in Tab. 7. We find that incorporating the language modality brings a modest improvement in performance (potentially because of inadequate language annotations), but it significantly increases the model parameters and GPU memory usage, and reduces the inference speed. One solution is to explore more efficient network architectures for underwater VL tracking.

## 6   Conclusion and Future Research

**Conclusion.** In this paper, we establish WebUOT-1M, *i.e.*, the first million-scale UOT dataset to facilitate the development of more powerful and versatile tracking systems. It is substantially larger and more diverse than existing UOT datasets, encompassing 1,500 video sequences across 408 object categories. The dataset covers various underwater scenarios and provides rich attributes and language prompts for comprehensive evaluation. Furthermore, a simple yet strong omni-knowledge distillation approach called OKTrack is proposed to boost the research of UOT. Evaluation of 30 deep trackers on WebUOT-1M reveals that Transformer-based and UOT-specific methods perform well. By providing a large-scale dataset, WebUOT-1M not only facilitates the evaluation and comparison of existing tracking algorithms but also paves the way for the development of new methodologies.

**Future Research.** Although WebUOT-1M significantly surpasses existing UOT datasets in terms of video sequences, target categories and underwater scenarios covered, our data size is still small compared to the latest multi-modal datasets, *e.g.*, LAION-5B [56] and InternVid [63]. In the future, we consider collecting more underwater videos, and building underwater datasets with more modalities, *e.g.*, depth and audio. By releasing the large-scale WebUOT-1M dataset, we hope it can inspire the community to develop large foundation models [76] for universal object tracking and broader fields, and broaden their application prospects [77].

**Acknowledgements.** This work was supported by the Guangzhou Municipal Science and Technology Project: Basic and Applied Basic research projects (No. 2024A04J4232), National Natural Science Foundation of China (No. 62101351), Guangzhou-HKUST(GZ) Joint Funding Program (Grant No.2023A03J0008), Education Bureau of Guangzhou Municipality, the Major Project of Technology Innovation and Application Development of Chongqing (CSTB2023TIAD-STX0015), and the Key Research and Development Program of Chongqing (cstc2021jscx-gksbX0032).

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
