# *Appendices* for WebUOT-1M: Advancing Deep Underwater Object Tracking with A Million-Scale Benchmark

**Chunhui Zhang**[1,2,3]**, Li Liu**[2]***Guanjie Huang**[2]**, Hao Wen**[3]**, Xi Zhou**[3]**, Yanfeng Wang**[1,4]

[1] Cooperative Medianet Innovation Center, Shanghai Jiao Tong University, Shanghai 200240, China
[2] The Hong Kong University of Science and Technology (Guangzhou), Guangzhou 511458, China
[3] CloudWalk Technology Co., Ltd, 201203, China
[4] Shanghai AI Laboratory, Shanghai 200232, China

## Appendices

Due to space concerns, many details have been omitted in the main text. Here, we present more details about our dataset and method, as well as discussions and experimental results, as follows.

- **Appendix A Social Impact.** We present the potential social impacts of our work.

- **Appendix B Limitations.** We discuss the limitations of our work.

- **Appendix C More Statistics about WebUOT-1M.** We offer more statistical results and dataset splits of WebUOT-1M.

- **Appendix D Details of Attributes.** We present the definitions and distributions of 23 tracking attributes.

- **Appendix E Details of Method.** We present more details about the proposed MATP module.

- **Appendix F Additional Discussions.** We perform extensive discussions and analyses on the sample imbalance, inference settings, the role of open-air domain knowledge, the advantages of the OKTrack method, and the differences between our work and existing works.

- **Appendix G Experiment Details.** We present more details of implementation and metrics.

- **Appendix H More Results.** We demonstrate the error ranges, results on UVOT400, and attribute-based performance on WebUOT-1M.

- **Appendix I Datasheet.** We provide the datasheet for WebUOT-1M.

## A   Social Impact

The proposed WebUOT-1M dataset can promote the research of UOT, which is beneficial for underwater vision understanding, marine environmental monitoring, marine animal conservation, *etc*. Despite our best efforts to collect as many target categories as possible, due to the vast diversity of underwater targets in the real world, we still need to be careful about whether models trained on WebUOT-1M can generalize well to *unseen* rare underwater targets. The constructed WebUOT-1M dataset under Creative Commons licenses 4.0[2] is intended solely for academic research purposes.

---

*Corresponding author. E-mail: avrillliu@hkust-gz.edu.cn.
[2]`https://creativecommons.org/licenses/`

## B Limitations

One limitation of the proposed method is that it relies on the ViT backbone, which is inherently constrained by the quadratic computational complexity of the self-attention mechanism [13]. It is interesting to explore more advanced architectures with linear computational complexity, *e.g.*, state space models [18, 48].

## C More Statistics about WebUOT-1M

We provide more statistics to help researchers fully understand and better use the proposed WebUOT-1M dataset. Figs. 1(a) and (b) present the distributes of the target position of training and test sets. The distribution of video size is demonstrated in Fig. 1(c). In Tab. 1, we compare the training and test sets of WebUOT-1M from multiple aspects, *e.g.*, the number of videos, the number of target categories, the total frames, and the total duration.

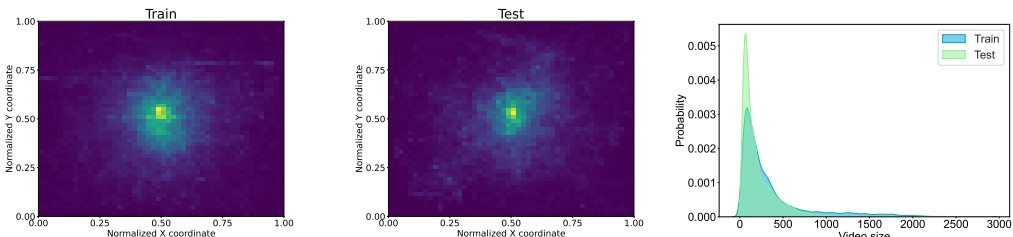

Figure 1: More statistics of WebUOT-1M. (a) Distribution of target center position of the training set. (b) Distribution of target center position of the test set. (c) Distribution of video size.

Table 1: Comparison between training and test sets of WebUOT-1M.

|  | Videos | Classes | Superclasses | Min frame | Mean frame | Max frame | Total frames | Total duration |
|---|---|---|---|---|---|---|---|---|
| WebUOT-1M test | 480 | 202 | 12 | 49 | 849 | 8,000 | 407 K | 3.86 hours |
| WebUOT-1M training | 1,020 | 287 | 12 | 49 | 680 | 9,985 | 693 K | 6.64 hours |
| WebUOT-1M | 1,500 | 408 | 12 | 49 | 733 | 9,985 | 1.1 M | 10.5 hours |

## D Details of Attributes

Tab. 2 demonstrates the definitions of 23 tracking attributes (low resolution (LR), fast motion (FM), scale variations (SV), aspect ratio variations (ARV), camera motion (CM), viewpoint changes (VC), partial occlusion (PO), full occlusion (FO), out-of-view (OV), rotation (ROT), deformation (DEF), similar distractors (SD), illumination variations (IV), motion blur (MB), partial target information (PTI), natural or artificial object (NAO), camouflage (CAM), underwater visibility (UV), watercolor variations (WCV), underwater scenarios (US), shooting perspective (SP), size (SIZ), and length (LEN) of video). We annotate 12 underwater scenarios, including sea, river, lake, pool, water tank, fish tank, basin, bowl, cup, aquarium, pond, and puddle. The WCV contains 16 watercolors (colorless, ash, green, light blue, gray, light green, deep blue, dark, gray-blue, partly blue, light yellow, light brown, blue, cyan, light purple, and blue-black). The distribution of attributes is shown in Fig. 2.

## E Details of Method

### E.1 Motion-aware Target Prediction

The Kalman filtering-based motion-aware target prediction (MATP) [23] is adopted to address tracking drift when the tracker incorrectly locates similar objects. For fast deployment, we borrowed the implementation from SORT [4] and UOSTrack [28]. Readers are strongly recommended to refer to [4] and [28] for more details. We summarize the workflow of MATP as follows:

Table 2: Descriptions of the 23 tracking attributes in WebUOT-1M.

| Attribute | Definition |
|---|---|
| 01. LR | If the size of the bounding box of the target in one frame is less than 400 pixels. |
| 02. FM | The center position of the target in two consecutive frames exceeds 20 pixels. |
| 03. SV | The ratio of the target bounding box is not within the range $[0.5, 2]$. |
| 04. ARV | The aspect ratio of the target bounding box is not in the range $[0.5, 2]$. |
| 05. CM | There is severe camera movement in the video frame. |
| 06. VC | Viewpoint changes significantly affect the appearance of the target. |
| 07. PO | If the target appears partially occluded in one frame. |
| 08. FO | As long as the target is completely occluded in one frame. |
| 09. OV | There is one frame where the target completely leaves the video frame. |
| 10. ROT | The target rotates in the video frame. |
| 11. DEF | The target appears deformation in the video frame. |
| 12. SD | Similarity interference appears around the target. |
| 13. IV | The illumination of the target area changes significantly. |
| 14. MB | The target area becomes blurred due to target motion or camera motion. |
| 15. PTI | In the initial frame only partial information about the target is visible. |
| 16. NAO | The target belongs to a natural or artificial object. |
| 17. CAM | The target is camouflaging in the video frame. |
| 18. UV | The underwater visibility of the target area (low, medium, or high visibility). |
| 19. WCV | The color of the water of the target area. |
| 20. US | Different underwater scenarios where the target is located. |
| 21. SP | Different shooting perspectives (underwater, outside-water, and fish-eye views). |
| 22. SIZ | The size $s = \sqrt{w \times h}$ of the video is small ($s < \sqrt{640 \times 480}$), medium ($\sqrt{640 \times 480} \leq s < \sqrt{1280 \times 720}$), or large ($s \geq \sqrt{1280 \times 720}$). |
| 23. LEN | The length $l$ of the video is short ($l \leq 600$ frames), medium ($600$ frames $< l \leq 1800$ frames), or long ($l > 1800$ frames). |

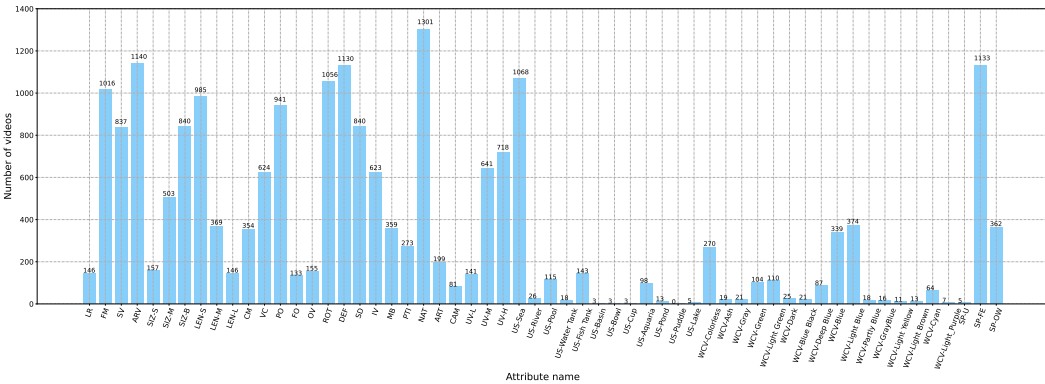

Figure 2: Distribution of videos in each attribute in WebUOT-1M. Best viewed by zooming in.

(1) **Candidate Set Extraction**: The search region is divided into $n \times n$ patches and the top-$N$ patches with the highest similarity scores to the template are extracted as a candidate set $C_t$.

(2) **Trajectory Prediction**: A Kalman filter ($kf$) is utilized to predict the target's position in the current frame, generating the estimation box $b^E$.

(3) **Location Score Calculation**: The location score between $b^E$ and each candidate box in $C_t$ is calculated using a combination of similarity score and IoU between the boxes.

(4) **Match Processing**: The tracker first predicts the target location using detection-based post-processing. If the IoU between the predicted box and $b^E$ is below a threshold, the tracker employs motion-based match processing. It calculates the location scores between $b^E$ and each candidate box in $C_t$ and outputs the candidate box with the highest score as the tracked target.

In summary, the MATP leverages trajectory prediction and matching to relocate the target hidden in candidate regions when tracking drift occurs. It effectively utilizes motion information and candidate

boxes in each frame, providing a new solution to improve tracking performance on similar object challenges for UOT and beyond. Note that MATP does not require training and is used directly during the inference phase. In Algorithm 1, we provide the pseudo-codes of the tracking model to conduct inference with MATP.

---

**Algorithm 1** Inference with MATP

---

**Input:** Kalman filter $kf$, first_frame, initial_box, response map, candidate set $C_t$, maximum response set $C'_t$, scores list $scores$, estimation box $b^E$, match result box $b^M$, IoU threshold $conf = 0.6$, response map threshold $threshold = 0.8$, IoU threshold $iou\_threshold = 0.5$, match state $match\_state = False$, *etc*

**Output:** Target boxes $B$

 1: $kf$.init(first_frame)
 2: $B = [initial\_box]$
 3: **for** $i = 2, 3, ..., T$ frames **do**
 4:     $C_t \leftarrow$ extract_candidates(response_map, $threshold$)
 5:     $C'_t \leftarrow$ NMS($C_t$)
 6:     $b^E \leftarrow kf$.predict()
 7:     $scores \leftarrow$ compute_scores($b^E, C'_t$)
 8:     **if** iou_of(max_response_box, $b^E$) $< conf$ **then**
 9:         $match\_state \leftarrow True$
10:     **else**
11:         $match\_state \leftarrow False$
12:     **end if**
13:     **if** $match\_state$ **then**
14:         $b^M \leftarrow$ argmax($scores$)
15:     **else**
16:         $b^M \leftarrow$ max_response_box
17:     **end if**
18:     $B$.append($b^M$)
19:     $kf$.update($b^M$)
20: **end for**
21: **return** $B$

---

## F    Additional Discussions

### F.1    Why is There A Sample Imbalance Between Underwater and Open-air Objects?

In the field of visual object tracking, commonly used open-air training data consists of approximately 20 M frames, including TrackingNet (14.43 M) [34], LaSOT (3.52 M) [15], GOT-10k (1.5 M) [21], and COCO (118 K) [30]. However, the previous largest underwater object tracking (UOT) dataset, *i.e.*, UVOT400, contains only 275 K frames. Considering the ratio of total frames between underwater and open-air datasets is **1:71**, we argue that there is a significant imbalance between underwater and open-air objects.

### F.2    Why Only Use Underwater Frames for Inference?

For underwater platforms, *e.g.*, unmanned underwater vehicles, the cameras typically deployed do not have image enhancement capabilities. Adding underwater image enhancement would result in additional energy consumption and latency for these low-power devices. Therefore, a reasonable and low-latency solution is to perform object tracking using only the underwater frames. Moreover, this also follows the evaluation of many existing UOT datasets [24, 35, 5, 1, 2]. As shown in Tab. 3, it is not surprising that tracking on enhanced frames can further improve performance. Therefore, developing lightweight and more effective underwater image enhancement algorithms is also a promising direction.

Table 3: Tracking using underwater frames *vs.* enhanced frames of OKTrack on WebUOT-1M.

| | Pre (%) | AUC (%) | nPre (%) | cAUC (%) | mACC (%) |
|---|---|---|---|---|---|
| Underwater frames | 57.5 | 60.0 | 63.8 | 59.3 | 61.0 |
| Enhanced frames | 58.1 (↑0.6) | 60.4 (↑0.4) | 64.5 (↑0.7) | 59.7 (↑0.4) | 61.3 (↑0.3) |

### F.3 Why is Open-air Domain Knowledge Useful for the UOT Task?

In our experiments, we used a teacher model pre-trained on large-scale open-air tracking datasets [15, 21, 34, 30, 29, 37, 43, 25] to guide the learning of the student tracker. For the learning of the student model, we utilized the proposed large-scale UOT dataset (WebUOT-1M). We argue that the student model needs to learn two primary abilities to achieve high performance in UOT: **general feature representation capability** and **domain-specific (*i.e.*, underwater environment) adaptive capability**. The WebUOT-1M dataset, with its rich variety of categories and comprehensive scene coverage, endows the student model with strong domain adaptation capabilities for the UOT task. Additionally, the open-air domain knowledge possessed by the teacher model (learned from large-scale open-air tracking datasets) is effectively imparted to the student model through the proposed omni-knowledge distillation. Given that current large-scale open-air tracking datasets have a more extensive data scale and cover more target categories and scenes compared to underwater datasets, they are beneficial for the student model to learn general feature representation capabilities. Therefore, open-air domain knowledge is very useful for the UOT task. In the future, we plan to continuously expand the scale of the established WebUOT-1M dataset and use more open-air datasets to further enhance the performance of UOT models.

### F.4 Why is Our Purely Visual Approach OKTrack Superior to the Current SOAT Visual Trackers and Vision-Language Tracking Methods?

In our experiments (see the main paper and Tab. 4), we were astonished to find that the proposed vision-based approach OKTrack surpassed SOTA visual trackers (*e.g.*, OSTrack [42], SeqTrack-B256 [7], and MixFormerV2-B [10]) even vision-language trackers (*e.g.*, CiteTracker-256 [26], All-in-One [45], and UVLTrack [31]). We speculate that this is due to several factors:

(1) **There is a significant gap between underwater and open-air domains.** Thus, directly applying existing open-air trackers (including visual trackers and vision-language trackers) to underwater environments leads to performance degradation. To quickly verify our hypothesis, we retrained the current SOTA visual tracker (OSTrack) and vision-language tracker (CiteTracker-256) using our WebUOT-1M dataset. The results (see the main paper and Tab. 4) show that retraining these open-air trackers on our underwater dataset indeed enhances their tracking performance. This is due to retraining reducing the gap between underwater and open domains. However, the proposed OKTrack still outperforms these retrained open-air trackers (both visual and vision-language trackers). *This can be attributed to the effectiveness of the proposed omni-knowledge distillation and MAPT.*

(2) **Limited vision-language tracking datasets.** The existing vision-language tracking datasets (including our WebUOT-1M) are still relatively small (in terms of language annotations), making it challenging to train or fine-tune large visual encoders. *It is a promising direction to construct larger-scale vision-language tracking datasets and benchmarks.*

(3) **Ambiguous language annotations.** Most language annotations in existing vision-language tracking datasets primarily describe the target's state in the initial frame. In long videos and some complex situations, these descriptions fail to accurately convey the target's current appearance changes. Therefore, training and testing models using the existing language annotations may mislead the models, resulting in decreased tracking performance. *A possible solution is to use existing multi-modal large models to generate more accurate language descriptions for the current vision-language tracking datasets, even to produce multi-granularity language descriptions (e.g., concise and detailed descriptions [27]) for a single video sequence.*

Table 4: Tracking using different prompts. We compare SOTA trackers using language prompt, language prompt+bounding box prompt, and bounding box prompt on WebUOT-1M. * denotes retraining on the WebUOT-1M training set.

| Method | Pre (%) | nPre (%) | AUC (%) | cAUC (%) | mACC (%) |
|---|---|---|---|---|---|
| Language prompt | | | | | |
| JointNLT [47] | 22.4 | 32.2 | 31.2 | 29.8 | 31.2 |
| UVLTrack [31] | 22.5 | 33.8 | 31.2 | 30.1 | 31.3 |
| Language prompt + bounding box | | | | | |
| JointNLT [47] | 25.5 | 34.9 | 32.7 | 31.5 | 32.8 |
| VLT$_{SCAR}$ [20] | 33.4 | 44.0 | 37.8 | 36.4 | 38.0 |
| VLT$_{TT}$ [20] | 41.7 | 52.1 | 48.3 | 47.3 | 48.8 |
| UVLTrack [31] | 52.5 | 60.0 | 55.8 | 55.0 | 56.6 |
| All-in-One [45] | 53.1 | 61.5 | 57.1 | 56.4 | 58.0 |
| CiteTracker-256 [26] | 49.3 | 58.4 | 54.6 | 53.7 | 55.2 |
| CiteTracker-256* [26] | 54.2 | 61.6 | 57.7 | 56.9 | 58.5 |
| Bounding box | | | | | |
| SeqTrack-B256 [7] | 50.8 | 58.5 | 54.0 | 53.3 | 54.7 |
| MixFormerV2-B [10] | 45.4 | 54.0 | 51.0 | 50.1 | 51.7 |
| OSTrack [42] | 52.9 | 61.1 | 56.5 | 55.8 | 57.4 |
| OSTrack* [42] | 55.1 | 61.3 | 57.0 | 56.2 | 57.8 |
| OKTrack (Ours) | 57.5 | 63.8 | 60.0 | 59.3 | 61.0 |

## F.5  Comparison with Previous Works

In this work, our primary contribution is the introduction of WebUOT-1M, *i.e.*, the largest and most diverse underwater tracking dataset in terms of target categories and underwater scenarios. Our dataset covers major underwater scenarios, target categories, and underwater instances in existing UOT datasets [24, 35, 2, 1] and open-air object tracking datasets [15, 14, 21]. The annotated tracking attributes include common attributes (*e.g.*, low resolution, fast motion, and illumination variations) as well as those specific to underwater scenes (*e.g.*, underwater visibility, watercolor variations, and camouflage). Based on the established WebUOT-1M dataset, we further propose a simple yet effective omni-knowledge distillation tracking framework, called OKTrack, for the community. The differences between our approach and existing works (*e.g.*, UVOT400 [1], HDETrack [38], UOSTrack [28]) are listed as follows:

- UVOT400 [1] introduced an underwater tracking dataset consisting of 400 video sequences, 275 K frames, and 17 different tracking attributes and target objects spanning 50 different categories. In comparison, our WebUOT-1M includes 1,500 video sequences, 1.1 million frames, 23 attributes, and 408 target categories. UVOT400 is a partially publicly available dataset, where only the annotations for the first frame of the test set are visible. The complete benchmark, source codes, and tracking results of our work, will be made publicly available.

- HDETrack [38] proposed a multi-modal knowledge distillation strategy for event-based tracking based on RGB frames and event streams. Drawing inspiration from this method, we present an omni-knowledge distillation framework for underwater tracking. In comparison to HDETrack, our approach exhibits several notable differences, such as the addition of token-based contrastive distillation loss and a motion-aware target prediction module.

- UOSTrack [28] presented a hybrid training strategy using both underwater images and open-air sequences to address sample imbalance, alongside employing motion-based post-processing to mitigate the influence of similar targets. We draw inspiration from its motion-based post-processing to address model drift caused by similar distractors, proposing a MATP module. Compared to UOSTrack, we contribute a new million-scale UOT dataset and demonstrate that training on such a dataset significantly enhances tracking performance, which will benefit the entire UOT community. Additionally, we introduce an omni-knowledge distillation framework for UOT.

## G  Experiment Details

### G.1  More Implementation Details

In our experimental evaluation, we only consider methods for which the code and model weights are publicly available. For fair comparisons, we use the code, weights, and default parameters provided by the original authors for evaluation. The experimental platform is an Ubuntu 20.04 server with two Intel(R) Xeon(R) Gold 6226R CPUs @ 2.90GHz, 8 NVIDIA A6000 GPUs and 512G Memory. Python 3.8.0 and PyTorch 2.0.1 are mainly used in our experiments.

For the proposed OKTrack, the template and the search region are $2^2$ times and $4^2$ times of the target bounding box, and then resized to $128 \times 128$ and $256 \times 256$, respectively. In underwater scenarios, using a larger template and search region may enhance tracking performance further, but it also entails increased computational costs. Additionally, due to the presence of dense similar distractors (*e.g.*, schools of fish) around target objects in underwater environments, enlarging the search region also increases the risk of model drift. Therefore, to reduce computational costs in underwater scenarios and achieve a fair comparison, for some SOTA trackers, we only consider their versions with a search region of $256 \times 256$, *e.g.*, OSTrack [42], SimTrack-B32 [6], MixFormerV2-B [10], SeqTrack-B256 [7], and CiteTracker-256 [26]. The teacher tracker [45] was trained using eight commonly used open-air tracking datasets (LaSOT [15], GOT-10k [21], TrackingNet [34], COCO [30], OTB99-L [29], TNL2K [37], WebUAV-3M [43], and VisualGenome [25]). In our experiments, we directly utilize the pre-trained weights of the teacher tracker. Following UOSTrack [28], two underwater object detection datasets (*i.e.*, RUOD [16], FishExtend [28]) are used to enhance the generalization of the tracking models.

### G.2  Metrics Details

Tab. 5 provides some descriptions of the adopted five evaluation metrics, *i.e.*, precision (Pre), normalized precision (nPre), success rate (AUC), complete success rate (cAUC), and mean accuracy (mACC). Readers are referred to [34, 15, 22, 43] for more details on each metric.

Table 5: Descriptions of five evaluation metrics.

| Metric | Description |
|---|---|
| **01. Pre** | The Pre is used to measure the percentage of frames where the center position error falls within a predefined threshold. Trackers are ranked based on this metric using a given precision score (*e.g.*, obtained when the threshold = 20 pixels). |
| **02. nPre** | As Pre is sensitive to target size and image resolution, nPre is introduced in [34], which normalizes each precision score over the size of the ground truth bounding box. |
| **03. AUC** | The AUC indicates the percentage of frames with overlap scores higher than a given threshold. Trackers are ranked based on this metric using the area under the curve (between 0 and 1) of each success plot. |
| **04. mACC** | The mACC measure proposed in [22], encourages trackers to provide reliable predictions for the target object even when it disappears. |
| **05. cAUC** | The above four metrics only measure center-point distance or overlap area and do not reflect the aspect ratio of the target object. To address this, [43] introduced the cAUC evaluation metric. Like the AUC, the cAUC is defined as the proportion of frames where the complete overlap score exceeds a specified threshold. |

## H  More Results

### H.1  Error Ranges

Following popular UOT and open-air tracking benchmarks [43, 15, 14, 37, 24, 35, 2, 1, 44], we perform the one-pass evaluation (OPE) for different tracking algorithms. To further verify the stability of different tracking algorithms, we conduct multiple tests using the Top-5 algorithms (*i.e.*, OKTrack, UOSTrack, All-in-One, GRM, OSTrack) on WebUOT-1M to obtain their error ranges. The results are shown in Tab. 6. We can see that these SOTA methods have very small fluctuations in performance across multiple tests.

Table 6: Error range of Top-5 trackers on WebUOT-1M.

| Method | Pre (%) | AUC (%) | nPre (%) | cAUC (%) | mACC (%) |
|---|---|---|---|---|---|
| OSTrack | $52.9_{\pm0.4}$ | $56.5_{\pm0.4}$ | $61.1_{\pm0.7}$ | $55.8_{\pm0.4}$ | $57.4_{\pm0.4}$ |
| GRM | $53.8_{\pm0.3}$ | $56.7_{\pm0.1}$ | $61.0_{\pm0.2}$ | $56.0_{\pm0.2}$ | $57.6_{\pm0.2}$ |
| All-in-One | $53.1_{\pm0.2}$ | $57.1_{\pm0.2}$ | $61.5_{\pm0.1}$ | $56.4_{\pm0.1}$ | $58.0_{\pm0.2}$ |
| UOSTrack | $54.3_{\pm0.3}$ | $58.3_{\pm0.2}$ | $62.6_{\pm0.1}$ | $57.5_{\pm0.2}$ | $59.1_{\pm0.2}$ |
| OKTrack | $57.5_{\pm0.1}$ | $60.0_{\pm0.1}$ | $63.8_{\pm0.1}$ | $59.3_{\pm0.2}$ | $61.0_{\pm0.2}$ |

## H.2 Results on UVOT400

To further validate the effectiveness of the proposed OKTrack, we show the tracking performance on the UVOT400 test set [1] in Tab. 7. Since the annotations for the UVOT400 test set are not visible, we submit the tracking results of UOSTrack and OKTrack to the official evaluation server to obtain AUC, nPre, and Pre scores. Results indicate that OKTrack achieves the best performance, with 63.2% in terms of AUC, 66.4% in terms of nPre, and 58.4% in terms of Pre. Compared to the previous best UOT tracker (UOSTrack), the gains of the proposed OKTrack are 1.8%, 1.6%, and 2.8% in terms of AUC, nPre, and Pre scores, respectively.

Table 7: Evaluation on UVOT400 test set. The reported results come from [1] or from our submissions to the official evaluation server.

| Method | AUC (%) | nPre (%) | Pre (%) |
|---|---|---|---|
| SiamFC [3] | 29.6 | 36.2 | 24.8 |
| SiamCAR [19] | 41.6 | 50.7 | 40.6 |
| PrDiMP [12] | 42.0 | 50.0 | 36.6 |
| ATOM [11] | 43.3 | 51.7 | 37.6 |
| STARK [41] | 43.4 | 49.9 | 40.4 |
| AutoMatch [46] | 48.6 | 59.8 | 47.0 |
| KeepTrack [33] | 49.4 | 59.0 | 44.1 |
| SiamBAN [9] | 49.8 | 61.2 | 47.6 |
| TransT [8] | 51.4 | 60.1 | 49.4 |
| TrDiMP [36] | 52.2 | 61.9 | 47.3 |
| ToMP-101 [32] | 53.9 | 63.7 | 51.4 |
| UOSTrack [28] | 61.4 | 64.8 | 55.6 |
| OKTrack (Ours) | 63.2 | 66.4 | 58.4 |

## H.3 Detailed Attribute-based Performance on WebUOT-1M

Fig. 3 shows the performance of 30 deep trackers on the WebOUT-1M test set of different attributes using **AUC** scores. Fig. 4 shows the performance of 30 deep trackers on the WebOUT-1M test set of different attributes using **Pre** scores. Fig. 5 shows the performance of 30 deep trackers on the WebOUT-1M test set of different attributes using **nPre** scores. Fig. 6 shows the performance of 30 deep trackers on the WebOUT-1M test set of different attributes using **cAUC** scores. We can observe that OKTrack achieves the best or comparable results across various attributes, indicating its robustness in facing a wide range of tracking challenges.

## H.4 Comparison with Multi-Object Tracking

We evaluate two SOTA multi-object frameworks for both single-object tracking and multi-object tracking (Unicorn [40] and MITS [39]) on four challenging underwater object tracking datasets (UOT100, UTB180, VMAT and WebUOT-1M) (see Tab. 8). The results demonstrate that our proposed OKTrack significantly outperforms both Unicorn and MITS.

# I Datasheet

Following [17], we provide the datasheet for our WebUOT-1M dataset:

Table 8: Comparison of our OKTrack and two SOTA multi-object tracking methods, *i.e.*, Unicorn and MITS. Pre/AUC scores are reported on UOT100, UTB180, VMAT and WebUOT-1M.

| Method | UOT100 | UTB180 | VMAT | WebUOT-1M |
|---|---|---|---|---|
| Unicorn [40] | 56.0/61.8 | 41.2/46.8 | 61.4/49.6 | 42.5/46.6 |
| MITS [39] | 60.8/64.9 | 56.2/59.8 | 64.1/50.1 | 52.1/54.1 |
| OKTrack (Ours) | 63.3/67.8 | 67.3/69.7 | 67.1/54.1 | 57.5/60.0 |

## I.1 Motivation

(1) **For what purpose was the dataset created?** Was there a specific task in mind? Was there a specific gap that needed to be filled? Please provide a description.

A1: The WebUOT-1M dataset was created to facilitate the development and evaluation of video-based underwater object tracking. 1) Previous UOT datasets suffer from limitations in scale, diversity of target categories, and scenarios covered, hindering the training and evaluation of modern tracking algorithms. 2) Existing UOT datasets only provide bounding box annotations, which do not support multi-modal underwater object tracking. 3) How to effectively transfer knowledge from large-scale open-air data to underwater tracking models has yet to be explored. To fill these gaps, we propose WebUOT-1M, *i.e.*, the largest and most diverse underwater tracking dataset in terms of target categories and underwater scenarios. Based on the established WebUOT-1M dataset, we further propose a simple yet effective omni-knowledge distillation tracking framework, called OKTrack, for the community. We believe that WebUOT-1M can contribute a valuable benchmark to the community for developing more general tracking models for UOT and broader fields.

(2) **Who created the dataset (*e.g.*, which team, research group) and on behalf of which entity (*e.g.*, company, institution, organization)?**

A2: This dataset was created by Chunhui Zhang, Li Liu, Guanjie Huang, Hao Wen, Xi Zhou, and Yanfeng Wang. Chunhui Zhang and Yanfeng Wang are from Shanghai Jiao Tong University, Li Liu and Guanjie Huang are from the Hong Kong University of Science and Technology (Guangzhou), and Hao Wen and Xi Zhou are from CloudWalk Technology. At the time of creation, Chunhui Zhang was a visiting Ph.D. student at the Hong Kong University of Science and Technology (Guangzhou).

(3) **Who funded the creation of the dataset?** If there is an associated grant, please provide the name of the grantor and the grant name and number.

A3: The project was funded by the National Natural Science Foundation of China (No. 62101351), and the Key Research and Development Program of Chongqing (cstc2021jscx-gksbX0032).

(4) **Any other comments?**

A4: None.

## I.2 Composition

(1) **What do the instances that comprise the dataset represent (*e.g.*, documents, photos, people, countries)?** Are there multiple types of instances (*e.g.*, movies, users, and ratings; people and interactions between them; nodes and edges)? Please provide a description.

A1: WebUOT-1M comprises 1.1 million frames with precise bounding box annotations across 1,500 underwater videos and 408 highly diverse target categories. These targets are further classed into 12 superclasses with reference to WordNet to facilitate the evaluation of the cross-superclass generalization ability of tracking models. We annotate the dataset with 23 tracking attributes and annotate a language prompt for each video.

(2) **How many instances are there in total (of each type, if appropriate)?**

A2: The dataset consists of 1,500 videos, totaling 1.1 million frames, and 10.5 hours.

(3) **Does the dataset contain all possible instances or is it a sample (not necessarily random) of instances from a larger set?** If the dataset is a sample, then what is the larger set? Is the sample representative of the larger set (*e.g.*, geographic coverage)? If so, please describe how this representativeness was validated/verified. If it is not representative of the larger set, please describe why not (*e.g.*, to cover a more diverse range of instances, because instances were withheld or unavailable).

A3: Despite our best efforts to collect as many target categories as possible, due to the vast diversity of underwater targets in the real world, we were unable to include all underwater targets in a single dataset. We will continue to increase the diversity and volume of WebUOT-1M in future work.

(4) **What data does each instance consist of?** "Raw" data (*e.g.*, unprocessed text or images) or features? In either case, please provide a description.

A4: Each instance includes diverse annotations (underwater images, bounding boxes, absent labels, the language prompt, the category name, and the superclass name).

(5) **Is there a label or target associated with each instance?** If so, please provide a description.

A5: Yes. See I.2 (1)-(4).

(6) **Is any information missing from individual instances?** If so, please provide a description, explaining why this information is missing (*e.g.*, because it was unavailable). This does not include intentionally removed information, but might include, *e.g.*, redacted text.

A6: No.

(7) **Are relationships between individual instances made explicit (*e.g.*, users' movie ratings, social network links)?** If so, please describe how these relationships are made explicit.

A7: Yes. We only annotate one instance per video and use a bounding box to represent their motion trajectory.

(8) **Are there recommended data splits (*e.g.*, training, development/validation, testing)?** If so, please provide a description of these splits, explaining the rationale behind them.

A8: The dataset is divided into a training set and a test set. Please refer to appendix C for details.

(9) **Are there any errors, sources of noise, or redundancies in the dataset?** If so, please provide a description.

A9: Despite our multiple rounds of careful annotation checks, there may still be some inaccuracies due to occlusion or blurring caused by the movement of underwater targets, such as slight shifts of bounding boxes. Manually annotated language prompts might not adequately describe the movement and appearance changes of targets over long video sequences. In the future, we plan to use large multi-modal models to further improve the accuracy and scientific quality of the language annotations.

(10) **Is the dataset self-contained, or does it link to or otherwise rely on external resources (*e.g.*, websites, tweets, other datasets)?** If it links to or relies on external resources, a) are there guarantees that they will exist, and remain constant, over time; b) are there official archival versions of the complete dataset (*i.e.*, including the external resources as they existed at the time the dataset was created); c) are there any restrictions (*e.g.*, licenses, fees) associated with any of the external resources that might apply to a dataset consumer? Please provide descriptions of all external resources and any restrictions associated with them, as well as links or other access points, as appropriate.

A10: Yes, the dataset is self-contained.

(11) **Does the dataset contain data that might be considered confidential (*e.g.*, data that is protected by legal privilege or by doctor–patient confidentiality, data that includes the content of individuals' non-public communications)?** If so, please provide a description.

A11: No.

(12) **Does the dataset contain data that, if viewed directly, might be offensive, insulting, threatening, or might otherwise cause anxiety?** If so, please describe why.

A12: No.

## I.3 Collection Process

(1) **How was the data associated with each instance acquired?** Was the data directly observable (*e.g.*, raw text, movie ratings), reported by subjects (*e.g.*, survey responses), or indirectly inferred/derived from other data (*e.g.*, part-of-speech tags, model-based guesses for age or language)? If the data was reported by subjects or indirectly inferred/derived from other data, was the data validated/verified? If so, please describe how.

A1: We assembled a professional annotation team from a qualified data labeling company. The author team conducted the last data verification to ensure high-quality annotations. Specifically,

in each frame of the video, the visual bounding box $[x, y, w, h]$ is used as the ground truth for the target, where $(x, y)$, $w$, and $h$ represent the target's top-left corner, width, and height, respectively. A sentence of language prompt describing the color, behavior, attributes, and surroundings of the target is given for each video sequence to encourage the exploration of multi-modal UOT.

(2) **What mechanisms or procedures were used to collect the data (*e.g.*, hardware apparatuses or sensors, manual human curation, software programs, software APIs)?** How were these mechanisms or procedures validated?

A2: Most of the video clips are collected from YouTube and BiliBili with careful filtering. We manually selected videos suitable for underwater object tracking and randomly chose targets to increase the diversity of our dataset. A professional data annotation team conducted multiple rounds of manual annotations, and the author team performed the final data verification to ensure high-quality annotations.

(3) **If the dataset is a sample from a larger set, what was the sampling strategy (*e.g.*, deterministic, probabilistic with specific sampling probabilities)?**
A3: N/A.

(4) **Who was involved in the data collection process (*e.g.*, students, crowdworkers, contractors) and how were they compensated (*e.g.*, how much were crowdworkers paid)?**
A4: The authors of the paper.

(5) **Over what timeframe was the data collected?** Does this timeframe match the creation timeframe of the data associated with the instances (*e.g.*, recent crawl of old news articles)? If not, please describe the timeframe in which the data associated with the instances was created.
A5: Collecting the data took about one month, and completing the data cleaning, organization, annotation, and verification took approximately six months.

(6) **Were any ethical review processes conducted (*e.g.*, by an institutional review board)?** If so, please provide a description of these review processes, including the outcomes, as well as a link or other access point to any supporting documentation.
A6: N/A.

### I.4 Preprocessing/cleaning/labeling

(1) **Was any preprocessing/cleaning/labeling of the data done (*e.g.*, discretization or bucketing, tokenization, part-of-speech tagging, SIFT feature extraction, removal of instances, processing of missing values)?** If so, please provide a description. If not, you may skip the remaining questions in this section.
A1: We manually collected and cleaned data from YouTube and BiliBili to ensure high-quality videos in WebUOT-1M. We discarded videos that are not suitable for tracking, such as repeated scenes, long-term static targets, and incomplete trajectories.

(2) **Was the "raw" data saved in addition to the preprocessed/cleaned/labeled data (*e.g.*, to support unanticipated future uses)?** If so, please provide a link or other access point to the "raw" data.
A2: No, we only provide the community with cleaned and annotated video sequences.

(3) **Is the software that was used to preprocess/clean/label the data available?** If so, please provide a link or other access point.
A3: We use the standard Python library Beautiful Soup to crawl videos from online websites. The dataset is manually annotated using an in-house annotation tool of the data labeling company.

(4) **Any other comments?**
A4: None.

### I.5 Uses

(1) **Has the dataset been used for any tasks already?** If so, please provide a description.
A1: No, this dataset is newly proposed.

(2) **Is there a repository that links to any or all papers or systems that use the dataset?** If so, please provide a link or other access point.
A2: N/A.

(3) **What (other) tasks could the dataset be used for?**

A3: WebUOT-1M can be used for underwater object tracking and underwater vision-language tracking. Additionally, it can be utilized for underwater vision understanding, marine environmental monitoring, and marine animal conservation.

(4) **Is there anything about the composition of the dataset or the way it was collected and preprocessed/cleaned/labeled that might impact future uses?** For example, is there anything that a dataset consumer might need to know to avoid uses that could result in unfair treatment of individuals or groups (*e.g.*, stereotyping, quality of service issues) or other risks or harms (*e.g.*, legal risks, financial harms)? If so, please provide a description. Is there anything a dataset consumer could do to mitigate these risks or harms?

A4: No.

(5) **Are there tasks for which the dataset should not be used?** If so, please provide a description.

A5: No.

(6) **Any other comments?**

A6: None.

## I.6  Distribution

(1) **Will the dataset be distributed to third parties outside of the entity (*e.g.*, company, institution, organization) on behalf of which the dataset was created?** If so, please provide a description.

A1: Yes, the WebUOT-1M dataset will be made publicly available to the community.

(2) **How will the dataset be distributed (*e.g.*, tarball on website, API, GitHub)?** Does the dataset have a digital object identifier (DOI)?

A2: The WebUOT-1M dataset will be publicly released on the GitHub project.

(3) **When will the dataset be distributed?**

A3: The WebUOT-1M dataset will be distributed once the paper is accepted after peer review.

(4) **Will the dataset be distributed under a copyright or other intellectual property (IP) license, and/or under applicable terms of use (ToU)?** If so, please describe this license and/or ToU, and provide a link or other access point to, or otherwise reproduce, any relevant licensing terms or ToU, as well as any fees associated with these restrictions.

A4: We release our dataset and benchmark under Creative Commons licenses 4.0.

(5) **Have any third parties imposed IP-based or other restrictions on the data associated with the instances?** If so, please describe these restrictions, and provide a link or other access point to, or otherwise reproduce, any relevant licensing terms, as well as any fees associated with these restrictions.

A5: No.

(6) **Do any export controls or other regulatory restrictions apply to the dataset or to individual instances?** If so, please describe these restrictions, and provide a link or other access point to, or otherwise reproduce, any supporting documentation.

A6: No.

(7) **Any other comments?**

A7: None.

## I.7  Maintenance

(1) **Who will be supporting/hosting/maintaining the dataset?**

A1: The authors of the paper.

(2) **How can the owner/curator/manager of the dataset be contacted (*e.g.*, email address)?**

A2: You can contact them via email on the GitHub project.

(3) **Is there an erratum? If so, please provide a link or other access point.**

A3: No.

(4) **Will the dataset be updated (*e.g.*, to correct labeling errors, add new instances, delete instances)?** If so, please describe how often, by whom, and how updates will be communicated to dataset consumers (*e.g.*, mailing list, GitHub)?

A4: To ensure the accuracy of the dataset, if the author identifies any errors or other researchers notify us of labeling errors in the data, we will promptly review and update the dataset.

(5) **If the dataset relates to people, are there applicable limits on the retention of the data associated with the instances (*e.g.*, were the individuals in question told that their data would be retained for a fixed period of time and then deleted)?** If so, please describe these limits and explain how they will be enforced.

A5: N/A.

(6) **Will older versions of the dataset continue to be supported/hosted/maintained?** If so, please describe how. If not, please describe how its obsolescence will be communicated to dataset consumers.

A6: No, we maintain the latest version of this dataset for the community on our GitHub project.

(7) **If others want to extend/augment/build on/contribute to the dataset, is there a mechanism for them to do so?** If so, please provide a description. Will these contributions be validated/verified? If so, please describe how. If not, why not? Is there a process for communicating/distributing these contributions to dataset consumers? If so, please provide a description.

A7: N/A. All researchers are welcome to collaboratively develop and extend WebUOT-1M.

(8) **Any other comments?**

A8: None.

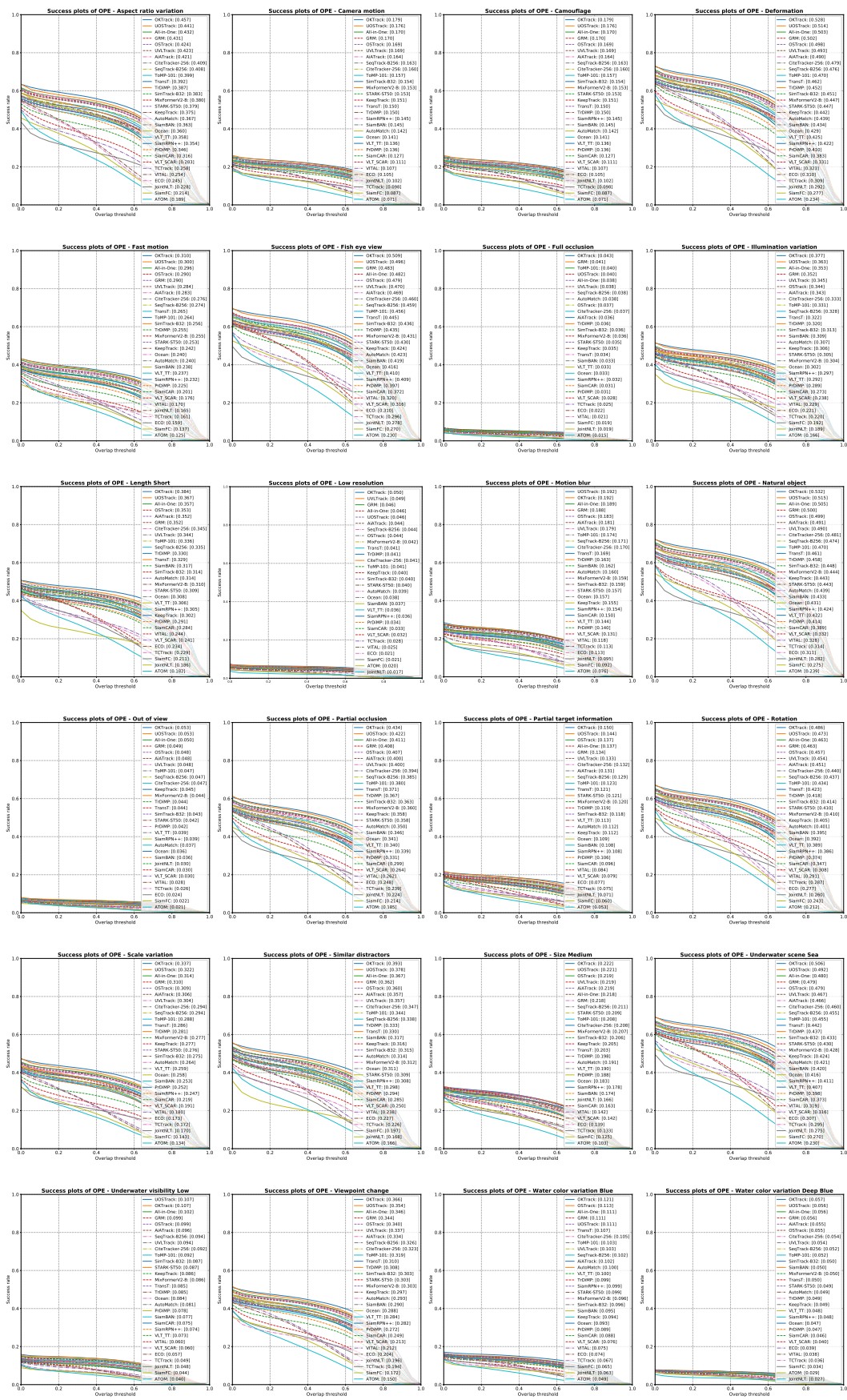

Figure 3: Performances of baseline trackers on the WebOUT-1M test set of different attributes using **AUC** scores. Best viewed by zooming in.

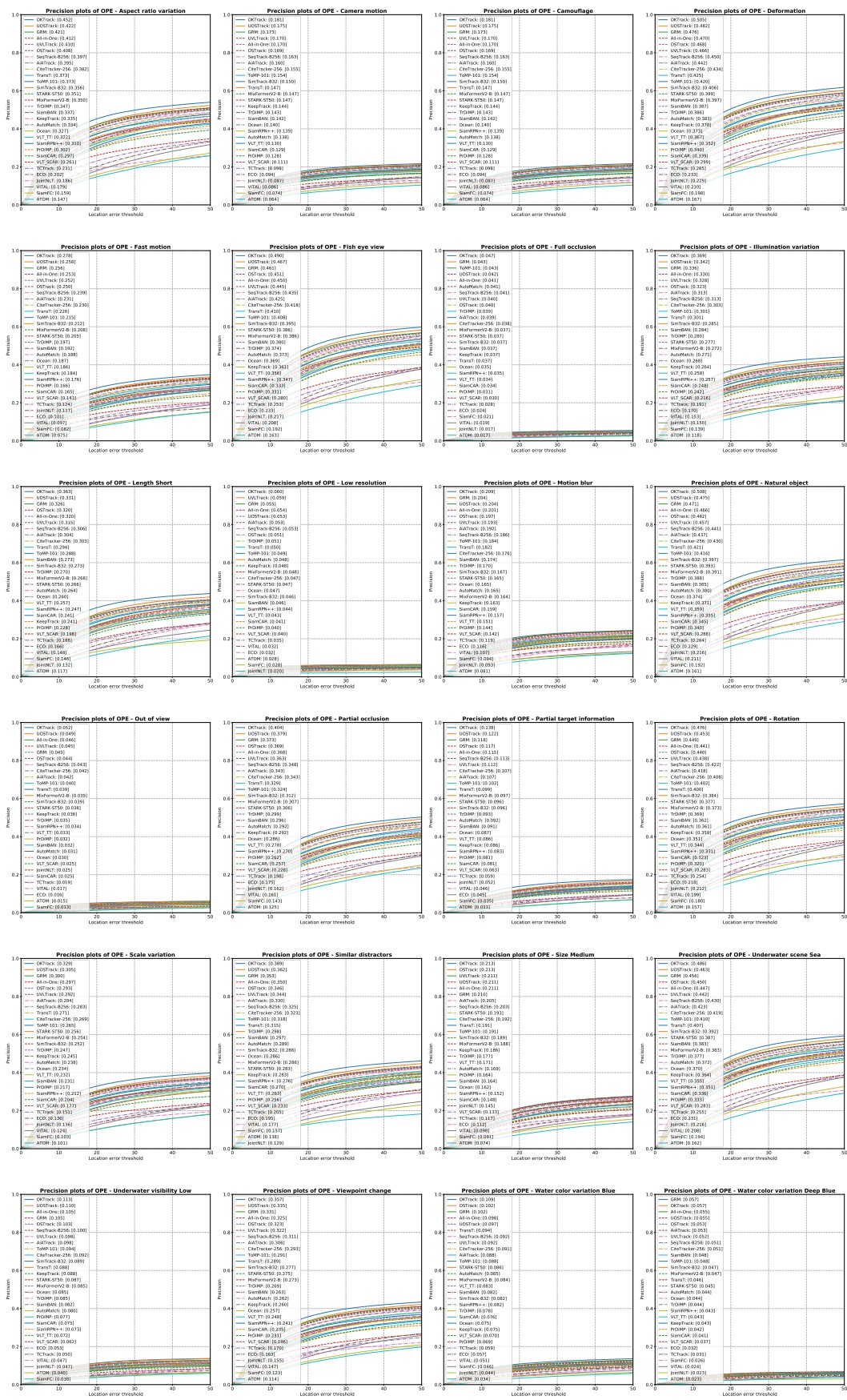

Figure 4: Performances of baseline trackers on the WebOUT-1M test set of different attributes using **Pre** scores. Best viewed by zooming in.

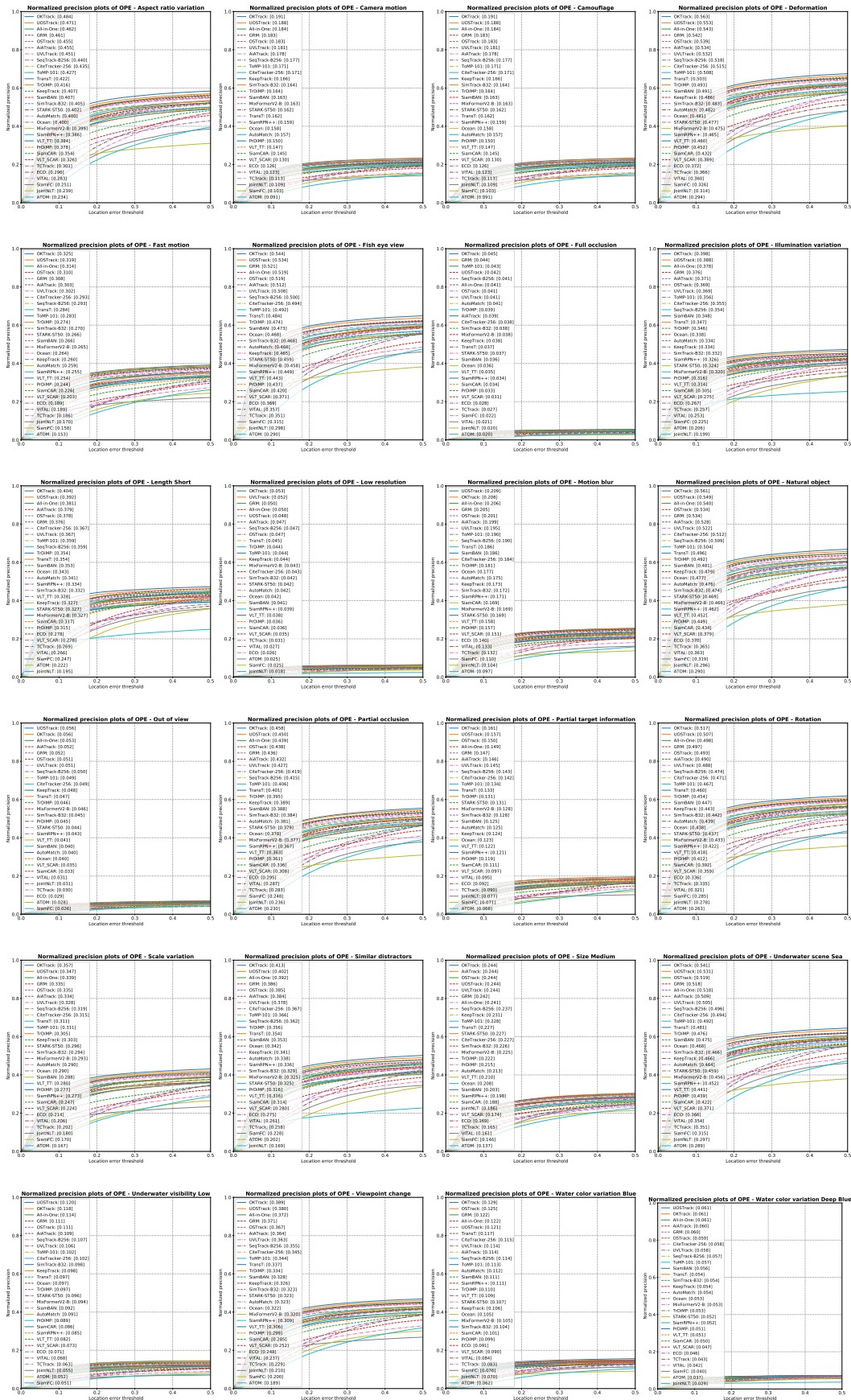

Figure 5: Performances of baseline trackers on the WebOUT-1M test set of different attributes using **nPre** scores. Best viewed by zooming in.

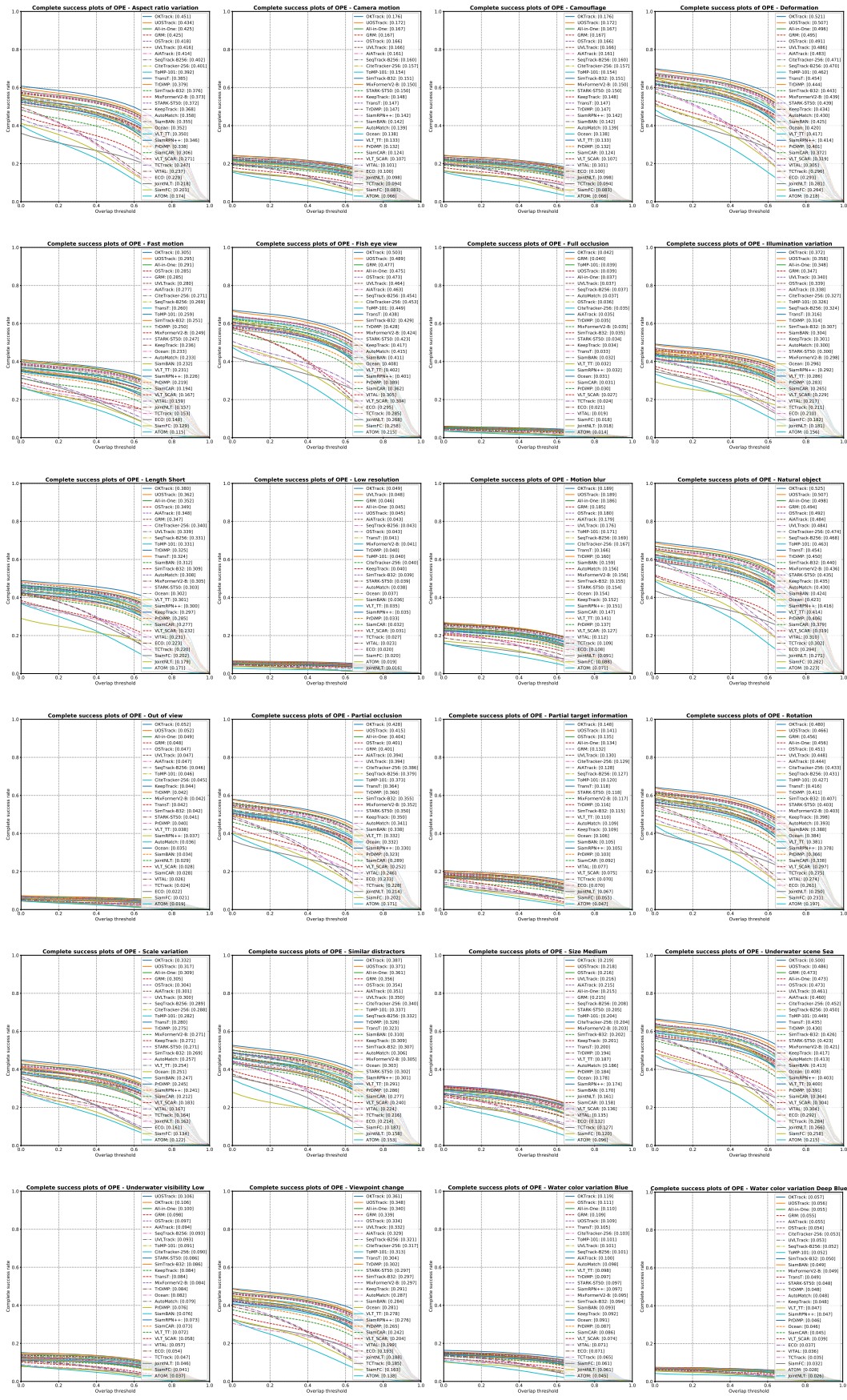

Figure 6: Performances of baseline trackers on the WebOUT-1M test set of different attributes using **cAUC** scores. Best viewed by zooming in.