# OpenReview forum: "WebUOT-1M: Advancing Deep Underwater Object Tracking with A Million-Scale Benchmark"
_NeurIPS.cc/2024/Datasets_and_Benchmarks_Track — NeurIPS 2024 Track Datasets and Benchmarks Poster_

### Official Review · Reviewer_7sq1 · 2024-07-14
**Review of WebUOT-1M**

**Rating:** 7
**Confidence:** 4
**Correctness:** Basically, the claims made in the sub…
**Clarity:** The written is satisfactory.

**Review:**

The first question is about the dataset category. WebUOT-1M contains 408 highly diverse target categories, which is surely more varied than previous datasets. However, as we all know, (single-object) VOT does not care about the object category. So what is the necessity for including such many categories?

My second question is about the dataset scale. As shown in Figure 4c, most videos of WebUOT-1M are used as the training dataset. However, the training of WebUOT-1M is not very effective. As shown in Table 3, take OSTrack [70] for example, the original Pre/AUC score is 52.9/56.5%, and the remaining result is 55.1/57.0%, where the improvement is small. From Table 2 we can see that OSTrack is not a UOT tracker, which is not trained on underwater data. So, my question is where do we need so much underwater data for training? Is the proposed large-scale UOT training data necessary?

An open question, the training on a large general tracking dataset and fine-tuning (domain adaption/generalization) on a specific dataset (e.g., underwater) is a viable and economical way, right? No need to re-establish a large training dataset (with full annotations) for each scene without essential differences (e.g., animals on the ground, underwater, or in the air).

Also, the GT annotation was also under the enhanced videos [33], which are the same as the proposed method. This may make the GT is in favor of the results of the proposed method. So, the enhanced frames can be also applied to the other trackers without retraining on WebUOT-1M, e.g., UOSTrack, All-in-One. Whether the performance can be improved?

The authors in the introduction claim that models trained on these early datasets struggle with unseen species, leading to poor generalization performance. However, the authors also do not provide the generalization ability of the proposed dataset, e.g., trained on WebUOT-1M and tested on other datasets.

The next question is about the method. As shown in Figure 5, the main difference between the teacher and student networks is the input of enhanced frames [33]. As shown, the objects in the enhanced frames are obviously clearer. A straightforward idea is to enhance the frames during training and inference before feeding into the network, without complex KD.

The word clouds shown in Figure 3 have many obvious mistakes that are not the underwater object categories. For example, ‘star’, ‘tiger’, ’red’, ‘yellow’, etc. The words are all double in the right figure, i.e., ‘swimming’, and ‘turtle’.

**Strengths:**

The proposed dataset is very large with various object categories.

The experiments provided in this work are abundant.

The writing of this manuscript is good.

**Additional Feedback:**

N/A

**Documentation:**

Basically sufficient.

**Ethics:**

I have no ethical concerns with the submission.

**Limitations:**

The authors addressed the limitations and potential negative societal impact of their work.

**Opportunities For Improvement:**

See the above 'Review' part.

**Relation To Prior Work:**

This work has discussed how this work differs from previous contributions.

**Summary And Contributions:**

This work builds WebUOT-1M, the largest public UOT benchmark to date.
WebUOT-1M comprises 1.1 million frames across 1,500 video clips filtered from 408 target categories, largely surpassing previous UOT datasets, e.g., UVOT400.
Additionally, WebUOT-1M includes language prompts for video sequences, expanding its application areas, e.g., underwater vision-language tracking.

---

> ### Author Rebuttal · Authors · 2024-08-19
>
> # Response to Reviewer 7sq1 [1/4]
>
> Dear Reviewer 7sq1,
>
> Thank you for your time and effort in reviewing our paper. We appreciate your recognition that the proposed **dataset is very large with various object categories**. We are also grateful for your acknowledgment that the **experiments provided in this work are abundant**, and that the **writing of this manuscript is good**. Your support is highly encouraging. In the following, we address your concerns about the weaknesses and the requested changes and we also revised the paper.
>
>
> ## Review:
>
> **1.Comment:** The first question is about the dataset category. WebUOT-1M contains 408 highly diverse target categories, which is surely more varied than previous datasets. However, as we all know, (single-object) VOT does not care about the object category. **So what is the necessity for including such many categories?**
>
> **Response:** Thanks for your comments. Our answer to your question is that it is necessary to construct a large-scale dataset with many categories. To address your concerns, we provide the following two explanations:
>
> - Firstly, we borrow the reasons given in VastTrack [A]："**To achieve general visual tracking like humans, the tracker is desired to “see” various sequences from an extremely large set of object categories during training to acquire the ability of generalization. However, the categories in existing large-scale datasets are rather limited.** For example, the popular TrackingNet and LaSOT comprise respectively 27 and 70 categories, which fall short for training universally generalizable trackers. Another popular dataset GOT-10k aims to handle this by largely expanding the number of object categories to 563. Despite its success in advancing generic-purpose tracking, the 563 object categories are still insufficient to represent massive diversity of categories present in the real world. **Besides training, a real general tracking system requires evaluation on videos of vast object categories, which can help mitigate biases to certain classes for more faithful assessment in real applications. However, the test sets of existing large-scale benchmarks all consist of less than 100 object categories, which may not be enough for a faithful assessment of general tracking.**" Therefore, from the perspectives of both model training and evaluation, we believe it is necessary to construct a large-scale dataset with many categories, especially for the Underwater Object Tracking scenario that this article is concerned with.
>
> - The points we propose in Appendix F.3 (lines 89-98): "We argue that the student model needs to learn two primary abilities to achieve high performance in UOT: **general feature representation capability** and **domain-specific (i.e., underwater environment) adaptive capability**. **The WebUOT-1M dataset, with its rich variety of categories and comprehensive scene coverage, endows the student model with strong domain adaptation capabilities for the UOT task.** Additionally, the open-air domain knowledge possessed by the teacher model (learned from large-scale open-air tracking datasets) is effectively imparted to the student model through the proposed omni-knowledge distillation. **Given that current large-scale open-air tracking datasets have a more extensive data scale and cover more target categories and scenes compared to underwater datasets, they are beneficial for the student model to learn general feature representation capabilities.**" Therefore, to learn the general feature representation capability and the domain-specific (i.e., underwater environment) adaptive capability, it is necessary to construct a large-scale dataset with many categories.
>
>
> [A] Liang Peng, Junyuan Gao, Xinran Liu, Weihong Li, Shaohua Dong, Zhipeng Zhang, Heng Fan, Libo Zhang. VastTrack: Vast Category Visual Object Tracking. arXiv 2024
>
> **2.Comment:** My second question is about the dataset scale. As shown in Figure 4c, most videos of WebUOT-1M are used as the training dataset. However, the training of WebUOT-1M is not very effective. As shown in Table 3, take OSTrack [70] for example, the original Pre/AUC score is 52.9/56.5%, and the retraining result is 55.1/57.0%, where the improvement is small. From Table 2 we can see that OSTrack is not a UOT tracker, which is not trained on underwater data. **So, my question is where do we need so much underwater data for training? Is the proposed large-scale UOT training data necessary?**
>
> **Response:** Thanks for your kindly suggestion. To address your concerns, we provide explanations from the following two aspects:
>
> **(1) Empirical Analysis on the Necessity of Large-scale UOT Training Data**
>
> - "The underwater environment usually exhibits uneven lighting conditions, low visibility, low contrast, watercolor variations, similar distractors, camouflage and target blurring, **posing distinct challenges for UOT** compared to traditional open-air tracking tasks [73, 19, 32, 62, 50]. Despite its significance, UOT has not been thoroughly explored due to the **absence of large-scale datasets, benchmarks**, and challenges in gathering abundant underwater videos [1,5]" (lines 32-38 of the main text). For the community, there is a lack of a large-scale UOT dataset for both the training and evaluation of modern tracking algorithms. Therefore, we need to construct large-scale UOT training data.
>
> - "Due to the huge appearance variation and behavioral differences among various marine animals, models trained on these **early (i.e., small-scale) datasets struggle with unseen species, leading to poor generalization performance**" (lines 43-45 of the main text). We argue that large-scale UOT training data can enhance the model's generalization ability (experimental results can be found in Comment 2 (2) and Comment 5). Thus, large-scale UOT training data is necessary.

---

> > ### Author Rebuttal · Authors · 2024-08-19
> >
> > # Response to Reviewer 7sq1 [2/4]
> > **(2) Experiments on the Necessity of Large-scale UOT Training Data**
> >
> > - From Tab. A, we observe that the original OSTrack was trained on **four large-scale open-air datasets**, thus acquiring good generalization capability. Even without training on the UOT dataset, it still achieved **52.9\%/56.5\%** in terms of Pre/AUC score on WebUOT-1M. However, when we trained OSTrack solely using **a relatively small-scale UOT dataset (i.e., UVOT400 training set)**, its performance significantly decreased (**from 52.9%/56.5% to 28.1\%/32.6\%** in terms of Pre/AUC score). When we trained OSTrack with the large-scale OUT dataset WebUOT-1M, we achieved **55.1%/57.0%** in terms of Pre/AUC score, surpassing the original OSTrack. The above results indicate that insufficient small-scale UOT training data make it difficult for the model to converge, let alone its generalization capability. **The tracking model can benefit from a large-scale UOT dataset, like WebUOT-1M; therefore, we argue that large-scale UOT training data is necessary.**
> >
> > Table A: Training OSTrack [70] using UVOT400 [1] and WebUOT-1M training sets. Pre/AUC scores are reported.
> >
> > | Method | Type | Training data | #Number of Videos/Images |  UTB180 | WebUOT-1M |
> > | :----:| :----: | :----: | :----: | :----: | :----: |
> > |OSTrack [70] |  Original   |  TrackingNet, LaSOT, GOT-10k, COCO  |  ~30,000+1120+9335 videos, 118,287 images   |  57.0/62.9 |   52.9/56.5  |
> > ||||||
> > |OSTrack [70] |  Retraining  | UVOT400 training set|  280 videos  | 30.0/35.2   |   28.1/32.6  |
> > |OSTrack [70] | Retraining | WebUOT-1M training set |  1020 videos  |  60.8/63.2  | 55.1/57.0 |
> >
> > - Furthermore, we explored the impact of different numbers of training videos on model performance using the proposed WebUOT-1M. The results are presented in Tab. B. We found that more UOT videos can enhance the model's performance. **Therefore, collecting a larger scale of UOT training data may further bring additional gains. This also proves that large-scale UOT training data is necessary.**
> >
> > Table B:  Training our OKTrack using different numbers of videos with WebUOT-1M. Pre/AUC scores are reported on UTB180 and WebUOT-1M.
> >
> > | Method | Rate |**#Number of Videos**   |  **UTB180**  | **WebUOT-1M**   |
> > | :----:| :----: | :----: | :----: | :----: |
> > | OKTrack (Ours)|  40\%   |  408  |  61.4/66.2 | 52.3/56.9 |
> > | OKTrack (Ours) |  60\%     | 612   | 62.8/67.0 |  53.7/57.6 |
> > | OKTrack (Ours)  |  80\% |  816   |  66.0/68.9 |  56.1/58.9 |
> > | OKTrack (Ours)  |  100\% |  1020  |  67.3/69.7  | 57.5/60.0 |
> >
> >
> > **3.Comment:** An **open question**, the training on a large general tracking dataset and fine-tuning (domain adaption/generalization) on a specific dataset (e.g., underwater) is a viable and economical way, right? **No need to re-establish a large training dataset (with full annotations) for each scene without essential differences (e.g., animals on the ground, underwater, or in the air).**
> >
> > **Response:** Thanks for your question. We answer your question as follows:
> >
> >
> > - We agree with your point: "The training on a large general tracking dataset and fine-tuning (domain adaptation/generalization) on a specific dataset (e.g., underwater) is a viable and economical approach."
> >
> > - However, we do not fully agree with your statement: "No need to re-establish a large training dataset (with full annotations) for each scene without essential differences (e.g., animals on the ground, underwater, or in the air)." The reason is that: **"The underwater environment usually exhibits uneven lighting conditions, low visibility, low contrast, watercolor variations, similar distractors, camouflage and target blurring, posing distinct challenges for UOT compared to traditional open-air tracking tasks [73, 19, 32, 62, 50]"** (lines 32-36 of the main text). Therefore, annotated training data for supervised learning of the tracking model is still necessary. Our experiments also demonstrate that using a large training dataset (e.g., WebUOT-1M) can enhance performance (see Tab. 3 of the main text).
> >
> > - **A compromise solution would be to construct a large-scale annotated UOT training dataset while also collecting massive high-quality unlabeled training data for domain adaptation/generalization as you suggested. We argue that constructing a fully annotated large-scale training dataset is valuable but costly, hence the need for a large-scale unlabeled training dataset to enable model learning.**

---

> > > ### Author Rebuttal · Authors · 2024-08-19
> > >
> > > # Response to Reviewer 7sq1 [3/4]
> > > **4.Comment:** Also, the GT annotation was also under the enhanced videos [33], which are the same as the proposed method. This may make the GT is in favor of the results of the proposed method. So, the enhanced frames can be also applied to the other trackers without retraining on (enhanced) WebUOT-1M, e.g., UOSTrack, All-in-One. Whether the performance can be improved?
> > >
> > > **Response:** Thanks for your constructive suggestions.
> > >
> > > - Following your suggestion, we use the same enhancement method [33] to obtain the enhanced WebUOT-1M test set. Then, we test two SOTA trackers (UOSTrack and All-in-One) on the enhanced WebUOT-1M test set without retraining on the enhanced WebUOT-1M. The results are shown in Tab. C.
> > >
> > > - **The results indicate that both algorithms (mainly trained on clear videos) have significant improvements in the enhanced video frames as you mentioned.** We speculate that the possible reasons are: on the one hand, enhanced video frames can provide more accurate and robust features; on the other hand, when trackers are trained on clear videos, they will have a preference for enhanced video frames during testing.
> > >
> > >
> > > Table C: Inference using underwater frames vs. enhanced frames of UOSTrack, and All-in-One on WebUOT-1M.
> > >
> > > |Method| WebUOT-1M test set|Pre (\%) | AUC (\%) | nPre (\%)  |
> > > | :----:| :----: | :----: | :----: | :----: |
> > > |UOSTrack  |Underwater frames | 54.3  | 58.3 |  62.6 |
> > > |UOSTrack  |Enhanced frames | 56.2  | 60.1  | 63.8  |
> > > ||||||||
> > > | All-in-One |Underwater frames | 53.1  | 57.1 |  61.5  |
> > > | All-in-One |Enhanced frames | 55.4   | 59.4   | 63.1   |
> > >
> > > [33] Shirui Huang, Keyan Wang, Huan Liu, Jun Chen, and Yunsong Li. Contrastive semi-supervised learning for underwater image restoration via reliable bank. In Proceedings of the IEEE/CVF conference on computer vision and pattern recognition, pages 18145–18155, 2023.
> > >
> > > **5.Comment:** The authors in the introduction claim that models trained on these early datasets struggle with unseen species, leading to poor generalization performance. However, the authors also do not provide the generalization ability of the proposed dataset, e.g., trained on WebUOT-1M and tested on other datasets.
> > >
> > > **Response:** Thanks for your comments. Actually, in addition to evaluating the performance of the proposed method OKTrack (trained on the WebUOT-1M training set) on the WebUOT-1M test set, we tested OKTrack on other datasets in Section 5.6 Empirical Discussions (see Fig. 10), including three widely used UOT datasets: UOT100, UTB180, and VMAT, to verify the generalization ability of the proposed dataset. We have the following analysis:
> > >
> > >
> > > - "**Comparison to Other UOT Benchmarks.** We experimentally compare WebUOT-1M with three open-source UOT datasets [52, 2, 5]. From Figs. 7 and 10, we obtain some valuable insights. **1) OKTrack achieves the best results on UTB180 and VMAT, and a comparable result on UOT100. The possible reason, as noted in [2], is that UOT100 contains a large amount of annotation errors.** 2) Compared with existing UOT datasets, WebUOT-1M is a more challenging and comprehensive benchmark suitable for both short-term tracking [52, 2] and long-term tracking [5]. 3) The relatively poor result on the long-term tracking dataset VMAT (see Fig. 10) indicates that long-term tracking is still challenging. One solution is to utilize the rich temporal information in video sequences" (lines of the main text).
> > >
> > > - **To further validate the effectiveness and generalizability of the proposed OKTrack, we have demonstrated the tracking performance on the UVOT400 test set [1] in Table 7 (see Appendix H.2 Results on UVOT400).** We stated: "Since the annotations for the UVOT400 test set are not visible,
> > > we submit the tracking results of UOSTrack and OKTrack to the offcial evaluation server to obtain
> > > AUC, nPre, and Pre scores. Results indicate that OKTrack achieves the best performance, with 63.2%
> > > in terms of AUC, 66.4% in terms of nPre, and 58.4\% in terms of Pre. Compared to the previous best
> > > UOT tracker (UOSTrack), the gains of the proposed OKTrack are 1.8\%, 1.6\%, and 2.8\% in terms of
> > > AUC, nPre, and Pre scores, respectively." (lines 195-200 of the Appendix)
> > >
> > > - **Overall, OKTrack achieves the best results on WebUOT-1M, UTB180, VMAT, and UVOT400, and a comparable result on UOT100, demonstrating the generalization ability of the proposed dataset (and OKTrack). For your convenience, we have displayed the results of OKTrack on UOT100, UTB180, VMAT, and UVOT400 in the attached PDF file.**

---

> > > > ### Author Rebuttal · Authors · 2024-08-19
> > > >
> > > > # Response to Reviewer 7sq1 [4/4]
> > > > **6.Comment:** The next question is about the method. As shown in Figure 5, the main difference between the teacher and student networks is the input of enhanced frames [33]. As shown, the objects in the enhanced frames are obviously clearer. A straightforward idea is to enhance the frames during training and inference before feeding into the network, without complex KD.
> > > >
> > > > **Response:** Thanks for your constructive suggestions.
> > > >
> > > > - Following your suggestion, we straightforwardly trained a baseline tracker (i.e., **OSTrack++**) of the proposed OKTrack using the enhanced WebUOT-1M training set and performed inference on the enhanced WebUOT-1M test set **without KD**. Results are shown in Tab. D. **It should be noted that our OKTrack uses OSTrack as the student network, without KD degrading into retraining OSTrack.**
> > > >
> > > > - From Tab. D, we the following observations: 1) The performance of the baseline tracker (OSTrack++), which was trained and tested on enhanced video frames, shows further improvement (56.3%/58.4%), as compared to the baseline tracker (OSTrack) that was trained and tested on underwater frames (55.1%/57.0%). 2) OSTrack exhibits improved performance on the enhanced video frames, with an increase from 55.1%/57.0% to 55.6%/57.5%. 3) However, there is a decrease in the performance of OSTrack++ when tested on underwater video frames, dropping from 56.3%/58.4% to 54.9%/56.7%. We hypothesize that this performance decline is due to OSTrack++ not having been exposed to underwater video frames with significant color deviation and blur during its training phase, resulting in reduced effectiveness when subsequently tested on such frames.
> > > >
> > > > - **Thank you once again for your constructive suggestions. In the future, we will explore simpler and more effective methods for underwater video enhancement to further improve the performance of UOT.**
> > > >
> > > >
> > > > Table D: Retraining on WebUOT-1M training set using underwater frames and enhanced frames. Pre/
> > > > AUC scores are reported on WebUOT-1M and enhanced WebUOT-1M test sets.
> > > >
> > > > | Method  | Training Data | WebUOT-1M test set|  Enhanced WebUOT-1M test set |
> > > > | :----:| :----: | :----: | :----: |
> > > > | Baseline (OSTrack)  | Underwater Frames |   55.1/57.0  |   55.6/57.5  |
> > > > |Baseline (OSTrack++)  | Enhanced Frames   |  54.9/56.7  | 56.3/58.4  |
> > > >
> > > > [33] Shirui Huang, Keyan Wang, Huan Liu, Jun Chen, and Yunsong Li. Contrastive semi-supervised learning for underwater image restoration via reliable bank. In Proceedings of the IEEE/CVF conference on computer vision and pattern recognition, pages 18145–18155, 2023.
> > > >
> > > > [75] Chunhui Zhang, Xin Sun, Yiqian Yang, Li Liu, Qiong Liu, Xi Zhou, and Yanfeng Wang. All in one: Exploring unified vision-language tracking with multi-modal alignment. In Proceedings of the 31st ACM International Conference on Multimedia, pages 5552–5561, 2023.
> > > >
> > > > **7.Comment:** The word clouds shown in Figure 3 have many obvious mistakes that are not the underwater object categories. For example, ‘star’, ‘tiger’, ’red’, ‘yellow’, etc. The words are all double in the right figure, i.e., ‘swimming’, and ‘turtle’.
> > > >
> > > > **Response:** We appreciate your constructive comments. We have carefully checked the object categories in the word cloud (Fig. 3) and found that the two problems you pointed out were caused by the word cloud library's tokenizer and our incorrect parameter setting for *repeat* and *collocations* in **WordCloud ()** function.
> > > >
> > > > - Due to the wrong tokenization of object category names, we get some undesirable words in Fig. 3. For example, "star" is from "east star zebrafish"; "tiger" is from "tiger angelfish", "tiger shark", or "sand tiger shark"; "red" is from "red tail phoenix"; "yellow" is from "yellow tang". **We have replaced the spaces in the object category names with underscores to ensure correct word segmentation**, e.g. "east_star_zebrafish", "tiger_angelfish", "tiger_shark", "sand_tiger_shark", and "yellow_tang".
> > > >
> > > > - We have set repetition and collocations parameters to **repeat=False** and **collocations=False** to **avoid repeated words in the word cloud**.
> > > >
> > > > - We have revised Fig. 3 and updated our paper accordingly. **For your convenience, we have included the updated Fig.3 in the attached PDF file.**
> > > >
> > > >
> > > > Thank you again for your helpful suggestions. We hope our response and the revised paper address your concerns, and we are happy to answer any further questions you may have.
> > > >
> > > >
> > > > Sincerely,
> > > >
> > > > Authors

---

> > > > > ### Comment · Reviewer_7sq1 · 2024-08-25
> > > > > **Comments after the rebuttal**
> > > > >
> > > > > First, thanks to the authors for their careful response to my comments.
> > > > >
> > > > > Regarding the concerns, for the original Comment 2, the empirical analysis can not address my question of the experimental results. In the experiment analysis part, the authors also claim that OSTrack trained on four large-scale open-air datasets can acquire good generalization capability. This way, why still need a large underwater one?
> > > > >
> > > > > Regarding the advantages, for the original Comments 4, and 6, the authors provide new detailed experimental results, which are even discrepant from the original points in the original paper. I appreciate the author being honest in providing the real results.
> > > > >
> > > > > Considering both the concerns, and the workload of this benchmark and the response, I am confused to judge.
> > > > >
> > > > > As the final evaluation, I am willing to support this work to be accepted, which is useful or not can be verified by more researchers in this community and even others, like the researchers on underwater life, in the future.

---

> > > > > > ### Author Rebuttal · Authors · 2024-08-28
> > > > > >
> > > > > > # Response to Reviewer 7sq1-Comment 2 [1/3]
> > > > > >
> > > > > > Dear Reviewer 7sq1,
> > > > > >
> > > > > > Thank you for your time and effort in reviewing our paper. We appreciate your affirmation and encouragement for our careful responses. We are also grateful that after your final evaluation, you are willing to support our work to be accepted. Your support is highly encouraging. In the following, we further address your concerns in Comment 2 from three perspectives: **average gains**, **scaling law of training data**, and **open discussions**.
> > > > > >
> > > > > >
> > > > > > **2.Comment:** My second question is about the dataset scale. As shown in Figure 4c, most videos of WebUOT-1M are used as the training dataset. However, the training of WebUOT-1M is not very effective. As shown in Table 3, take OSTrack [70] for example, the original Pre/AUC score is 52.9/56.5%, and the retraining result is 55.1/57.0%, where the improvement is small. From Table 2 we can see that OSTrack is not a UOT tracker, which is not trained on underwater data. **So, my question is where do we need so much underwater data for training? Is the proposed large-scale UOT training data necessary?**
> > > > > >
> > > > > > **Response:** Thanks for your kindly suggestions. To address your concerns, we provide further explanations from the following three perspectives:
> > > > > >
> > > > > > **(1) Average Gains of Large-scale UOT Training Data**
> > > > > >
> > > > > > - **We note that the reviewer observed that the improvement of OStrack [70] is small (e.g., the original Pre/AUC score is 52.9/56.5%, and the retraining result is 55.1/57.0% on WebUOT-1M), and therefore questioned why a large underwater dataset is still needed.** Actually, as shown in Tab. 3, OSTrack has still achieved certain gains, 2.2% and 0.5% in terms of Pre and AUC scores, respectively.
> > > > > >
> > > > > > - The reasons for OSTrack's relatively small improvement are multifaceted: for instance, although we trained the model with the same parameters as the original paper [70], these may not be optimal for underwater scenarios. **However, we have noticed that other retrained trackers show very significant improvements (see Tab. 3 in the main text or Tab. A below).**
> > > > > >
> > > > > > - **From a different view, we want to assess whether a large underwater dataset is still needed based on average gains. We believe that the average gains may be more reliable than a single metric of one tracker.** From Tab. B, we observe average gains of **5.6%/3.5%** and **3.2%/4.0%** in terms of Pre/AUC scores on UTB180 and WebUOT-1M for four representative trackers (ATOM, SiamFC, OSTrack, CiteTracker). **These significant average gains indicate that a large UOT tracking dataset can substantially enhance the performance of trackers from an overall perspective.** Moreover, the retrained trackers use less data compared to the original trackers (WebUOT-1M vs. TrackingNet, LaSOT, GOT-10k, and COCO).
> > > > > >
> > > > > >
> > > > > > Table A:  Performance comparison of original and retraining with WebUOT-1M training set. Pre/AUC scores are reported. (from Tab. 3 of the main text)
> > > > > >
> > > > > > | Method |   UTB180 (original)  | UTB180 (retraining)    |   WebUOT-1M (original)  | WebUOT-1M (retraining)   |
> > > > > > | :----:| :----: | :----: | :----: |  :----: |
> > > > > > | ATOM|    19.3/31.4 |25.7/36.5 |18.9/27.0 |21.4/32.6
> > > > > > | SiamFC |    22.3/35.1| 27.5/39.1 |22.5/31.6| 25.8/38.2|
> > > > > > | OSTrack  |    57.0/62.9 | 60.8/63.2 | 52.9/56.5 | 55.1/57.0 |
> > > > > > | CiteTracker-256  |   54.5/61.7| 61.6/66.3 |49.3/54.6| 54.2/57.7|
> > > > > > | **Avg. performance**  | 38.3/47.8  | 43.9/51.3   | 35.9/42.4 | 39.1/46.4 |
> > > > > > | **Avg. gains**  |  -  |  **5.6/3.5**  |   -  |   **3.2/4.0**   |
> > > > > >
> > > > > >
> > > > > > **(2) Scaling Law of Large-scale UOT Training Data**
> > > > > >
> > > > > > - **Motivation:** The existing large-scale open-air tracking datasets (e.g., LaSOT, GOT-10k, and TrackingNet) have brought great recent advances in open-air tracking. However, current UOT datasets suffer from **limitations in scale**, diversity of target categories and scenarios covered, **hindering the training and evaluation of modern tracking algorithms**. This inspired us to construct a **large-scale UOT dataset (i.e., WebUOT-1M)** with diverse target categories and multimodal annotations (e.g., bounding boxes and language prompts) to **facilitate both training and evaluation of trackers in this area**.
> > > > > >
> > > > > > - **Background of scaling law**
> > > > > >   - The scaling law [A, B], in the context of artificial intelligence and machine learning, refers to a principle that describes the relationship between the performance of a large model and three key factors: computational resources, the number of model parameters, and **the amount of training data**. According to this principle, the ultimate performance of a large model is primarily determined by these three factors, rather than the specific architecture of the model, such as the number of layers, depth, or width.
> > > > > >
> > > > > >   - The core idea of the scaling law is that there exists a power-law relationship between model performance and these three variables (i.e., computational resources, the number of model parameters, and the amount of training data). This means that to improve model performance, it is usually necessary to increase the number of model parameters and **the size of the data**.

---

> > > > > > > ### Author Rebuttal · Authors · 2024-08-28
> > > > > > >
> > > > > > > # Response to Reviewer 7sq1-Comment 2 [2/3]
> > > > > > > - **Scaling law of training data**
> > > > > > >
> > > > > > >   - **Our goal is to leverage the scaling law to explain why there is still a need for a large-scale UOT dataset.**
> > > > > > >
> > > > > > >   - First, we add two groups of experiments (see Tab. B) to compare the performance of retraining OSTrack using open-air datasets [18, 50, 32, 43] and underwater datasets [1, C]. **The results demonstrate that using a larger amount of training data from both open-air datasets and underwater datasets can improve model performance, which is in accordance with the scaling law principle.**
> > > > > > >
> > > > > > >   - Then, let's display the results from Tab B. in the form of a line chart (**see Fig. 1 in the attached PDF file**). Fig. 1 clearly demonstrates the **emergence of the scaling-up phenomenon in the proposed large-scale UOT training data** (i.e., more extensive training data leads to better performance). **We believe that the scaling law supports the necessity of large UOT training data. Thus, the large-scale UOT training data is necessary to some degree.**
> > > > > > >
> > > > > > > Table B: Performance comparison of training OSTrack using open-air datasets [18, 50, 32, 43] and underwater datasets [1, C]. Pre/AUC scores are reported.
> > > > > > >
> > > > > > > | Method | Type | Training Data | Number of Videos |  UTB180 | WebUOT-1M |
> > > > > > > | :----:| :----: | :----: | :----: | :----: | :----: |
> > > > > > > |OSTrack |  Retraining   |  Open-air data [18] |  280 videos   |  24.1/28.5 |   21.5/26.3  |
> > > > > > > |OSTrack |  Retraining   |  Open-air data [18] |  1120 videos   |  44.2/53.0 |   41.4/47.1  |
> > > > > > > |OSTrack |  Original   |  Open-air data [18, 50, 32, 43] |  ~40 K videos  |  57.0/62.9 |   52.9/56.5  |
> > > > > > > ||||||
> > > > > > > |OSTrack |  Retraining  | Underwater data [1] |  280 videos  | 30.0/35.2   |   28.1/32.6  |
> > > > > > > |OSTrack | Retraining | Underwater data [C] |  1020 videos |  60.8/63.2  | 55.1/57.0 |
> > > > > > >
> > > > > > > [A]  Kaplan J, McCandlish S, Henighan T, et al. Scaling laws for neural language models. arXiv preprint arXiv:2001.08361, 2020.
> > > > > > >
> > > > > > > [B] Alabdulmohsin I M, Neyshabur B, Zhai X. Revisiting neural scaling laws in language and vision. Advances in Neural Information Processing Systems, 2022, 35: 22300-22312.
> > > > > > >
> > > > > > > [C] Chunhui Zhang, Li Liu, Guanjie Huang, Hao Wen, Xi Zhou, Yanfeng Wang. WebUOT-1M: Advancing Deep Underwater Object Tracking with A Million-Scale Benchmark. arXiv preprint arXiv:2405.19818, 2024.
> > > > > > >
> > > > > > > [1] Basit Alawode, Fayaz Ali Dharejo, Mehnaz Ummar, Yuhang Guo, Arif Mahmood, Naoufel Werghi, Fahad Shahbaz Khan, and Sajid Javed. Improving underwater visual tracking with a large-scale dataset and image enhancement. arXiv preprint arXiv:2308.15816, 2023.
> > > > > > >
> > > > > > > [18] Heng Fan, Hexin Bai, Liting Lin, and et al.. Lasot: A high-quality large-scale single object tracking benchmark. International journal of computer vision, 129(2):439–461, 2021.
> > > > > > >
> > > > > > > [32] Lianghua Huang, Xin Zhao, and Kaiqi Huang. Got-10k: A large high-diversity benchmark for generic object tracking in the wild. IEEE Transactions on Pattern Analysis and Machine Intelligence, 43(5):1562– 428
> > > > > > > 1577, 2019.
> > > > > > >
> > > > > > > [43] Tsung-Yi Lin, Michael Maire, Serge Belongie, and et al.. Microsoft coco: Common objects in context.
> > > > > > > European Conference on Computer Vision, pages 740–755, 2014.
> > > > > > >
> > > > > > > [50] Matthias Muller, Adel Bibi, Silvio Giancola, Salman Alsubaihi, and Bernard Ghanem. Trackingnet: A large-scale dataset and benchmark for object tracking in the wild. In European Conference on Computer Vision, pages 300–317, 2018.
> > > > > > >
> > > > > > > - Furthermore, we explore the impact of different numbers of training videos on model performance using the proposed WebUOT-1M. The results are presented in Tab. B. We find that more UOT videos can enhance the model's performance. **Therefore, collecting a larger scale of UOT training data may further bring additional gains. This also proves that large-scale UOT training data is necessary.**
> > > > > > >
> > > > > > > Table B:  Training OKTrack using different numbers of videos with WebUOT-1M. Pre/AUC scores are reported.
> > > > > > >
> > > > > > > | Method | Rate |**Number of Videos**   |  **UTB180**  | **WebUOT-1M**   |
> > > > > > > | :----:| :----: | :----: | :----: | :----: |
> > > > > > > | OKTrack (Ours)|  40\%   |  408  |  61.4/66.2 | 52.3/56.9 |
> > > > > > > | OKTrack (Ours) |  60\%     | 612   | 62.8/67.0 |  53.7/57.6 |
> > > > > > > | OKTrack (Ours)  |  80\% |  816   |  66.0/68.9 |  56.1/58.9 |
> > > > > > > | OKTrack (Ours)  |  100\% |  1020  |  67.3/69.7  | 57.5/60.0 |

---

> > > > > > > > ### Author Rebuttal · Authors · 2024-08-28
> > > > > > > >
> > > > > > > > # Response to Reviewer 7sq1-Comment 2 [3/3]
> > > > > > > > **(3) Open Discussions on the Necessity of Large-scale UOT Training Data**
> > > > > > > >
> > > > > > > > - **Underwater data is critical for a variety of applications, and the need for large training data like the proposed large-scale WebUOT-1M training dataset can be justified for several reasons:**
> > > > > > > >
> > > > > > > >   - **Research and Development:** Advancing technologies in robotics, navigation, and computer vision often require large amounts of data to improve algorithms and models. Underwater data can help in developing more sophisticated algorithms for underwater object detection, tracking, and scene understanding.
> > > > > > > >
> > > > > > > >   - **Marine Conservation:** Understanding marine ecosystems is crucial for conservation efforts. Underwater data can be used to monitor the behavior of marine species, track the health of coral reefs, and observe the impacts of climate change.
> > > > > > > >
> > > > > > > >   - **Military and Defense:** Submarine operations and mine detection require detailed knowledge of the underwater environment. Training data can be used to improve sonar systems and autonomous underwater vehicles (AUVs).
> > > > > > > >
> > > > > > > >   - **Practical Applications:** In practical applications such as underwater monitoring, search and rescue, or military operations, even **small improvements in tracking performance can have significant operational benefits.**
> > > > > > > >
> > > > > > > > - **Here’s why the proposed large-scale underwater object tracking training data might be necessary:**
> > > > > > > >
> > > > > > > >   - **Complex Environmental Conditions:** Underwater environments are challenging due to factors like uneven lighting conditions, low visibility, low contrast, watercolor variations, similar distractors, camouflage, and target blurring. Large datasets can help models learn to handle these conditions effectively.
> > > > > > > >
> > > > > > > >   - **Diverse Object Categories:** There is a wide variety of objects and creatures in underwater settings, and a large dataset can ensure that models are trained to recognize and track a broad spectrum of them.
> > > > > > > >
> > > > > > > >   - **Handling of Edge Cases:** **Open-air datasets may not cover the full range of scenarios and edge cases that are common in underwater environments.** A dedicated dataset can ensure that these are accounted for in training.
> > > > > > > >
> > > > > > > >   - **Continuous Learning:** Underwater environments are dynamic, and having access to extensive training data allows for continuous learning and model updates to adapt to new conditions or objects.
> > > > > > > >
> > > > > > > >   - **Domain Adaptation:** The generalization capability and robustness of OSTrack might be good enough for a baseline, but performing effective domain adaptation (such as knowledge distillation) on the model with an underwater dataset could lead to better adaptation to underwater frames, potentially improving accuracy and robustness, like the proposed OKTrack.
> > > > > > > >
> > > > > > > >   - **Algorithmic Advances:** Having a large underwater training dataset can inspire the development of new algorithms or modifications to existing ones that are specifically optimized for underwater tracking, potentially leading to breakthroughs that wouldn’t be possible with generalist approaches.
> > > > > > > >
> > > > > > > >   - **Benchmarking:** Large datasets serve as benchmarks to compare the performance of different algorithms and to push the boundaries of what is currently possible in underwater object tracking.
> > > > > > > >
> > > > > > > >
> > > > > > > > Thank you again for your time and helpful suggestions, which enable us to have a deeper understanding of this field. We hope our responses can address your concerns.
> > > > > > > >
> > > > > > > > Sincerely,
> > > > > > > >
> > > > > > > > Authors

---

> > > > > > > > > ### Comment · Reviewer_7sq1 · 2024-09-02
> > > > > > > > > **Final discussion**
> > > > > > > > >
> > > > > > > > > Many thanks for the effort from the authors. I hope this benchmark can bring value to our community.

---

### Official Review · Reviewer_9mKy · 2024-07-23
**Review of WebUOT-1M**

**Rating:** 6
**Confidence:** 4
**Correctness:** Yes
**Clarity:** Yes.

**Review:**

Pros:
1. The proposed dataset is large-scale. Compared with other datasets, it has many advantages, as shown in Table 1.
2. Many baseline methods are evaluated on WebUOT-1M dataset. It is helpful to other researchers in the community.

Cons:
1. There is an ambiguity in the experiment. In Table 3 and Table 5, retraining OKTrack outperforms the distillation method shown in Table 5. However, Fig. 9 shows that the distillation outperforms retraining. It seems not reasonable.
2. In Fig. 2, we can see a lot of missing bounding boxes for animals. If there is a crowd of fish, we can see only a few objects are annotated. It is not reasonable.
3. The dataset should also contain multi-object tracking performance comparison.
4. Table 1 does not provide the number of annotated boxes for comparison.

**Strengths:**

1. The dataset is large-scale, which is helpful for the underwater perception community.
2. Experiments are conducted on the proposed dataset, which can be used as a benchmark.

**Additional Feedback:**

No

**Documentation:**

Yes

**Limitations:**

Yes

**Opportunities For Improvement:**

Please see the cons in the review.

**Relation To Prior Work:**

Yes.

**Summary And Contributions:**

In this paper, the authors propose the WebUOT-1M, a million-scale benchmark dataset featuring diverse underwater video sequences. In addition, the authors propose OKTrack, which combines different distillation approaches for underwater tracking. Besides, several baseline methods are compared in the WebUOT-1M dataset.

---

> ### Author Rebuttal · Authors · 2024-08-19
>
> # Response to Reviewer 9mKy [1/2]
>
> Dear Reviewer 9mKy,
>
> We thank you for your time and effort spent reviewing our paper. We are glad that you recognize the **advantages of our WebUOT-1M dataset** compared to other datasets, as well as the **comprehensive experimental evaluation** on our dataset which is **helpful to other researchers in the community**. We also appreciate your suggestions for our paper. We have updated our paper based on your comments.
>
>
> ## Review:
> **1.Comment:** There is an ambiguity in the experiment. In Table 3 and Table 5, retraining OKTrack outperforms the distillation method shown in Table 5. However, Fig. 9 shows that the distillation outperforms retraining. It seems not reasonable.
>
> **Response:** Sorry for the unclear description. **Tab. 3 is from Section 5.4 Evaluation Results**, while **Tab. 5 and Fig. 9 are from Section 5.5 Ablation Study**. In this work, the training epochs for the experiments in Section 5.4 Evaluation Results and Section 5.5 Ablation Study are different. Therefore, their results cannot be directly compared.
>
> - As stated in Section 5.1 Implementation Details: **"The batch size and total epoch are 32 and 300" (for our final baseline tracker OKTrack in Tab. 3 from Section 5.4 Evaluation Results)**; **"For proper and fast verification, models are trained for 50 epochs in ablation experiments" (for trackers in Tab. 5 and Fig. 9 from Section 5.5 Ablation Study)**.
>
> - In Tab. 3, there is no original tracker for OKTrack, as our OKTrack is trained from scratch using WebUOT-1M. **Please do not compare it from the training strategy perspective with the methods in Tab.5 and Fig.9, as their training setting (epochs) are different.**
>
> - For Tab.3 and Tab.5, our final tracker (OKTrack trained for 300 epochs in Tab. 3) outperforms the tracker from the ablation analysis (OKTrack trained for 50 epochs in Tab. 5). This is because a larger number of training epochs allows the model to be trained more thoroughly and stably.
>
> - "In Fig. 9, we compare three different training settings: retraining the student tracker using WebUOT-1M and open-air tracking datasets, fine-tuning the student tracker, and adopting omni-knowledge distillation for the student tracker on WebUOT-1M. The omni-knowledge distillation achieves the best performance. Fine-tuning the model is preferable to retraining it, as the former can mitigate the issue of insufficient data to some extent, while the latter is limited by the sample imbalance between underwater and open-air objects" (lines 297-302 of the main text). The conclusion is: Retraining < Fine-tuning < Omni-Knowledge Distillation.
>
> **We apologize again for any confusion caused by our unclear expression. We will continue to refine our paper to make it more readable.**
>
>
> **2.Comment:** In Fig. 2, we can see a lot of missing bounding boxes for animals. If there is a crowd of fish, we can see only a few objects are annotated. It is not reasonable.
>
> **Response:** Thanks for your kindly comments.
>
> - Underwater object tracking is a **single-object tracking task**, where we only need to track one moving target from each video. Therefore, for each video in the UOT task, bounding box annotations are provided for only one moving target. To facilitate the understanding of the UOT task for the readers, we have provided the definition of UOT: **"Underwater object tracking (UOT) refers to the task of sequentially locating a submerged instance in an underwater video, given its initial position in the first frame [1][5][37][52]"** (lines 26-28 of the main text).
>
> - In practice, you can annotate two fish from the same video. However, they will be named as two different video sequences to distinguish two different trajectories, such as fish-1 and fish-2. They are considered to be two different video sequences of the dataset, stored in two different files with the same video frames, but different bounding box annotations. During inference, for the given video sequence, we only need to track one target with the bounding box provided in the initial frame.
>
>
> [1] Basit Alawode, Fayaz Ali Dharejo, Mehnaz Ummar, Yuhang Guo, Arif Mahmood, Naoufel Werghi, 347
> Fahad Shahbaz Khan, and Sajid Javed. Improving underwater visual tracking with a large scale dataset and image enhancement. arXiv preprint arXiv:2308.15816, 2023.
>
> [5] Levi Cai, Nathan E McGuire, Roger Hanlon, T Aran Mooney, and Yogesh Girdhar. Semi-supervised visual tracking of marine animals using autonomous underwater vehicles. International Journal of Computer Vision, 131(6):1406–1427, 2023.
>
> [37] Landry Kezebou, Victor Oludare, Karen Panetta, and Sos S Agaian. Underwater object tracking benchmark and dataset. In 2019 IEEE International Symposium on Technologies for Homeland Security (HST), pages 1–6. IEEE, 2019.
>
> [52] Karen Panetta, Landry Kezebou, Victor Oludare, and Sos Agaian. Comprehensive underwater object tracking benchmark dataset and underwater image enhancement with gan. IEEE Journal of Oceanic Engineering, 47(1):59–75, 2021.

---

> > ### Author Rebuttal · Authors · 2024-08-19
> >
> > # Response to Reviewer 9mKy [2/2]
> > **3.Comment:** The dataset should also contain multi-object tracking performance comparison.
> >
> > **Response:** Thanks for your constructive suggestion. Following your suggestion, we evaluated **two SOTA multi-object tracking methods (Unicorn [A] and MITS [B])** on four challenging underwater object tracking datasets (UOT100, UTB180, VMAT and WebUOT-1M) (see Tab. A). The results demonstrate that our proposed OKTrack significantly outperforms both Unicorn and MITS. We will provide a detailed comparative analysis of multi-object trackers in our revised version.
> >
> >
> > Table A: Comparison of our OKTrack and two SOTA multi-object tracking methods, i.e., Unicorn [A] and MITS [B]. Pre/AUC scores are reported on UOT100, UTB180, VMAT and WebUOT-1M.
> >
> > | Method | UOT100 | UTB180 |  VMAT | WebUOT-1M |
> > | :----| :----: | :----: | :----: | :----: |
> > |Unicorn [A]| 56.0/61.8 |  41.2/46.8  |  61.4/49.6  |  42.5/46.6 |
> > | MITS [B] | 60.8/64.9 |  56.2/59.8  |  64.1/50.1  |  52.1/54.1 |
> > | | | | | |
> > |OKTrack (Ours) | 63.3/67.8 |  67.3/69.7  |  67.1/54.1  |  57.5/60.0 |
> >
> >
> >
> > **[A] Unicorn: Bin Yan, Yi Jiang, Peize Sun, Dong Wang, Zehuan Yuan, Ping Luo, Huchuan Lu.
> > "Unicorn: Towards Grand Unification of Object Tracking." ECCV 2022**
> >
> > **[B] MITS: Yuanyou Xu, Zongxin Yang, Yi Yang. "Integrating Boxes and Masks: A Multi-Object Framework for Unified Visual Tracking and Segmentation." ICCV 2023**
> >
> > **4.Comment:** Table 1 does not provide the number of annotated boxes for comparison.
> >
> > **Response:** Thanks for your kindly comment. In underwater object tracking datasets, only one bounding box is annotated on each frame (where the target does not exist, the bounding box annotation is [0,0,0,0]), so the **number of bounding boxes** is equal to the **Total frames** in Tab.1. Following your suggestion, we have added a column for the number of annotated boxes in Tab. 1 in our revised version.
> >
> > Thank you again for reviewing our paper. We hope our response and the revised paper address your concerns, and we are happy to answer any further questions you may have.
> >
> >
> > Sincerely,
> >
> > Authors

---

### Official Review · Reviewer_iASE · 2024-07-24
**A paper contributes a valuable underwater object tracking dataset**

**Rating:** 7
**Confidence:** 4
**Correctness:** Yes, the claims in this paper are cor…
**Clarity:** Yes, the writing of the paper is easy…

**Review:**

Pros.
* This paper contributes a large underwater object-tracking dataset, which is valuable for further studies in this research direction.
* The dataset provides high-quality annotations which will be useful to train object tracking models.
* The paper is well-written and easy-to-follow.

Cons.
* The source code is currently unavailable.
* Upright bounding box annotations may inaccurate for locating tragets.

**Strengths:**

* The paper constructs the first million-scale UOT dataset, substantially surpassing previous UOT datasets in size and scope.
* The proposed methods are thoroughly evaluated on the newly introduced WebUOT-1M and existing UOT datasets. The evaluation includes representative tracking algorithms based on CNN, CNN-Transformer, and Transformer architectures.

**Additional Feedback:**

See the comments above.

**Documentation:**

Although the paper provides sufficient details about the dataset, the proposed method's source code is currently unavailable.

**Ethics:**

I do not think there are ethical concerns in this paper.

**Limitations:**

The authors have included a comprehensive discussion on limitations.

**Opportunities For Improvement:**

* The ablation study results indicate that the number of Transformer layers significantly impacts performance. Is this the primary factor affecting performance?
* In other Transformer-based methods, does the number of Transformer layers also significantly impact performance?
* Minior issues: The abstract can made more compact.

**Relation To Prior Work:**

Yes, this paper discussed the difference between this work and the existing papers.

**Summary And Contributions:**

This paper addresses the limitations of current underwater object tracking (UOT) datasets in terms of scale and diversity. It introduces WebUOT-1M, a dataset comprising 1.1 million frames extracted from 1,500 video clips across 408 target categories. Additionally, the paper comprehensively evaluates WebUOT-1M using 30 different deep trackers. Furthermore, the paper proposes an omni-knowledge distillation framework based on WebUOT-1M, designed to transfer knowledge from open-air domains to UOT models.

---

> ### Author Rebuttal · Authors · 2024-08-19
>
> # Response to Reviewer iASE [1/2]
>
> Dear Reviewer iASE,
>
> We thank you for your time and effort spent reviewing our paper. We are glad that you recognize our paper's **contribution of a valuable underwater object-tracking dataset** for further studies with **high-quality annotations** for model training, and its **well-written and easy-to-follow** nature. We also appreciate your suggestions for our paper. We are accelerating the process of open-sourcing the complete dataset, source codes, and tracking results based on your suggestions. To promote future research, they will be available on our [Github homepage](https://github.com/983632847/Awesome-Multimodal-Object-Tracking) once the paper is accepted after peer review, as promised in our **Abstract and Appendices I.Datasheet**.
>
> ## Review:
>
> **1.Comment:** The source code is currently unavailable.
>
> **Response:**  Thanks for your kindly comment. The source codes will be available at [here](https://github.com/983632847/Awesome-Multimodal-Object-Tracking) after peer review.
>
> **2.Comment:** Upright bounding box annotations may be inaccurate for locating targets.
>
> **Response:** Thanks for your suggestion.
>
> - We agree with you that bounding box annotations may be inaccurate for locating targets. One solution is to provide segmentation masks for the interested targets. However, during our data annotation process, we found that even bounding box annotations are very challenging due to some underwater videos with severe color deviation and blurring. To ensure the quality of the annotations, we hired *"a professional data labeling team to conduct multiple rounds of manual annotation and correction. The author team performs the final data verification to ensure the high quality of the annotations"* (lines 129-133 of the main text). The data cleaning, organization,
> annotation, and verifcation took approximately six months.
>
> - The recently open-sourced **SAM 2** [A] can handle both images and videos to obtain satisfactory segmentation masks. In the future, we plan to try annotating segmentation masks for our WebUOT-1M to promote broader applications and research, such as underwater video object segmentation [B].
>
> [A] Ravi N, Gabeur V, Hu Y T, et al. Sam 2: Segment anything in images and videos[J]. arXiv preprint arXiv:2408.00714, 2024.
>
> [B] Lian S, Li H. Evaluation of Segment Anything Model 2: The Role of SAM2 in the Underwater Environment[J]. arXiv preprint arXiv:2408.02924, 2024.
>
>
> ## Opportunities For Improvement:
> **3.Comment:** The ablation study results indicate that the number of Transformer layers significantly impacts performance. Is this the primary factor affecting performance?
>
> **Response:** Sorry for the missing explanation.
>
> - Following [63], in our omni-knowledge distillation framework, both the teacher model and the student model are based on a ViT-based backbone with a maximum of 12 layers. The main difference is that the teacher model is pre-trained, while the student model requires training from scratch. In our ablation study (Tab. 6), we set both the teacher and student models to have the same number of transformer layers. For the teacher model, having more transformer layers means that it is more powerful. **Therefore, in our setup, the capability of the teacher model (i.e., the number of transformer layers) is a primary factor that significantly affects the performance of the student model (i.e., OKTrack) as you mentioned.**
>
> - **One insight we can derive is that attempting to use a better teacher model to enhance the performance of the student model during the knowledge distillation.** We sincerely thank the reviewer for your meticulous observation that have led us to such a valuable insight. Thank you so much!
>
>
> [63] Xiao Wang, Shiao Wang, Chuanming Tang, Lin Zhu, Bo Jiang, Yonghong Tian, and Jin Tang. Event stream-based visual object tracking: A high-resolution benchmark dataset and a novel baseline. CVPR 2024

---

> > ### Author Rebuttal · Authors · 2024-08-19
> >
> > # Response to Reviewer iASE [2/2]
> > **4.Comment:** In other Transformer-based methods, does the number of Transformer layers also significantly impact performance?
> >
> > **Response:** Thanks for your kindly suggestions.
> >
> > - Following your suggestion, we conducted a thorough investigation of the transformer-based trackers involved in our paper. We found that many methods did not conduct experimental analysis on the impact of the number of Transformer layers. Among the works that analyzed the impact of Transformer layers, **there are mainly two distinct conclusions: one is that Transformer layers have a slight impact on performance, and the other is that Transformer layers significantly affect performance.** Here we provide two examples: HDETrack [63] (see Tab. A) and OSTrack [70] (see Tab. B). For more details, please refer to the original papers [63], [70].
> >
> > - Here is our analysis: **The two significant differences between HDETrack and OSTrack are whether they employ knowledge distillation for model training and the scale of the training data. OSTrack does not use knowledge distillation but instead uses the ViT-Base model pre-trained with MAE as the backbone and is trained on four large-scale training datasets (COCO, TrackingNet, LaSOT, GOT-10k). Consequently, the number of Transformer layers has a smaller impact on the model's performance. In contrast, HDETrack and our OKTrack utilize knowledge distillation, where the performance of the student model is strongly dependent on the capability of the teacher model. Moreover, compared to large-scale open-air tracking datasets, the relatively smaller RGBE tracking dataset (EventVOT [63]) and UOT dataset (WebUOT-1M) make the student model more prefer to a strong teacher model (i.e., more Transformer layers). In other words, when the data is insufficient, a stronger teacher model is needed to achieve better performance.**
> >
> >
> > Table A: Ablation studies on the number of Transformer layers on COESOT dataset (Table 8 from HDETrack [63]).
> >
> > | #(COESOT). Number of Former Layers |SR |PR |NPR|
> > | :----:| :----: | :----: | :----: |
> > |12 layers | 52.3 |59.0| 58.0|
> > |8 layers |49.2| 55.5| 54.8|
> > |4 layers |42.1 |46.0 |46.3|
> >
> > Table B: Ablation studies on the number of encoder layers used for relation modeling on LaSOT (Table A4 from OSTrack [70]).
> >
> > | #Number of Transformer Layers |Success |P_Norm| P|
> > | :----:| :----: | :----: | :----: |
> > |12  | 68.7 |78.1| 74.6|
> > |6 |67.9 |77.3| 73.6|
> > |3 |67.8 |77.0 |73.5|
> >
> >
> > [63] Xiao Wang, Shiao Wang, Chuanming Tang, Lin Zhu, Bo Jiang, Yonghong Tian, and Jin Tang. Event stream-based visual object tracking: A high-resolution benchmark dataset and a novel baseline. CVPR 2024
> >
> > [70] Botao Ye, Hong Chang, Bingpeng Ma, Shiguang Shan, and Xilin Chen. Joint feature learning and relation modeling for tracking: A one-stream framework. ECCV 2022
> >
> >
> > **5.Comment:** Minor issues: The abstract can made more compact.
> >
> > **Response:** Thanks for your constructive suggestions. Here is a simplified version of our abstract:
> >
> > - Underwater Object Tracking (UOT) is essential for identifying and tracking submerged objects in underwater videos, but existing datasets are limited in size and variety, impeding the development of advanced tracking algorithms. To address this, we introduce WebUOT-1M, the largest public UOT dataset to date, featuring 1.1 million frames from 1,500 videos across 408 categories, significantly outperforming previous datasets like UVOT400. Through meticulous manual annotation and verification, we provide high-quality bounding boxes for underwater targets. Additionally, WebUOT-1M includes language prompts for video sequences, expanding its application areas, \eg, underwater vision-language tracking. Given that most existing trackers are designed for open-air conditions and perform poorly in underwater environments due to domain differences, we propose a novel framework that uses omni-knowledge distillation to train a student Transformer model effectively. To the best of our knowledge, this framework is the first to effectively transfer open-air domain knowledge to the UOT model through knowledge distillation, as demonstrated by results on both existing UOT datasets and the newly proposed WebUOT-1M. We have thoroughly tested WebUOT-1M with 30 deep trackers, proving its potential as a benchmark for future UOT research. The dataset, along with codes and tracking results, will be publicly accessible at [here](https://github.com/983632847/Awesome-Multimodal-Object-Tracking).
> >
> > - **We will continue to polish the abstract in our revised version.**
> >
> > Thank you again for reviewing our paper. We hope our response and the revised paper address your concerns, and we are happy to answer any further questions you may have.
> >
> >
> > Sincerely,
> >
> > Authors

---

### Official Review · Reviewer_Fqcu · 2024-07-27
**A new large scale tracking datasets under the water setting**

**Rating:** 6
**Confidence:** 5

**Review:**

Quality: The paper is of high quality, presenting a well-organized and thoughtful approach to constructing a new underwater object tracking (UOT) benchmark. The authors have clearly demonstrated their expertise in the field and have provided a detailed methodology for the construction of WebUOT-1M.

Clarity: The paper is well-written and easy to understand, with clear explanations of the proposed methods and benchmarks. The authors have effectively communicated the purpose and contributions of their work.

Originality: The work presents a significant original contribution to the field of UOT. The authors have successfully addressed the limitations of existing UOT benchmarks by proposing a new approach to generate diverse text annotations using LLMs.

Significance: The significance of this work cannot be overstated. WebUOT-1M has the potential to drive substantial advancements in UOT research by providing a more nuanced and challenging dataset for algorithm development and evaluation. The proposed method, OKTrack, has the potential to enhance the performance of UOT algorithms by providing a more representative and diverse dataset.

**Strengths:**

Diverse Benchmark: WebUOT-1M offers a diverse set of underwater video sequences, which provides a comprehensive evaluation of UOT algorithms across different scenarios.

Experimental Analysis: The authors have conducted comprehensive experimental analyses on WebUOT-1M, evaluating the impact of diverse text on tracking performance.

**Additional Feedback:**

none.

**Clarity:**

The paper is well-written and presents the work in a clear and organized manner. The authors have effectively communicated the purpose, methodology, and results of their research. The following aspects contribute to the clarity of the paper:

Introduction and Motivation:
The introduction provides a clear rationale for the need for a new UOT benchmark and outlines the limitations of existing benchmarks, motivating the proposed work.
Methodology:
The methodology section describes the construction of WebUOT-1M, making it easy for readers to understand the process.

Claims and Evidence:
The authors’ claims are supported by evidence from the experiments conducted, and the evidence is clearly explained in the paper.
Discussion of Limitations:
The authors openly discuss the limitations of their work, which adds to the clarity by setting realistic expectations about the dataset and its potential impact.

**Correctness:**

The claims made in the submission are generally correct, and the evaluation methods and experiment design appear to be appropriate and correctly performed. However, there are a few points that could be further clarified to ensure the correctness of the claims and the evaluation process as mentioned above.

**Documentation:**

none.

**Ethics:**

none.

**Limitations:**

The authors have addressed some limitations in the paper, but there are areas that could be further developed to enhance the robustness and applicability of their work. Here are two specific limitations that could be considered for improvement:

Overlap with Existing Work:
The baseline tracker proposed in the paper shares a high degree of similarity with the tracker proposed in EventVOT (CVPR 2024). To ensure that the authors’ work is distinct and original, they should either provide a more detailed explanation of how their baseline tracker differs from EventVOT or present new contributions that significantly advance the state of the art beyond what is already known.

Constructive Suggestion: The authors could provide a comparative analysis of their baseline tracker with EventVOT, highlighting unique aspects or improvements that their approach brings to the field. This would help to establish the novelty and significance of their work.

Integration of Language for Tracking:
While the paper proposes tracking using language, the baseline tracker is not developed based on this setting. This inconsistency between the proposed method and the baseline tracker could undermine the overall contribution of the paper. To ensure that the paper’s approach is cohesive and its results are meaningful, the authors should either integrate language into the baseline tracker or justify why the baseline tracker is used without language integration.


Constructive Suggestion: The authors could either revise the baseline tracker to include language processing or provide a clear explanation of why language processing is not suitable for their baseline tracker. This would help to clarify the relationship between the proposed method and the baseline tracker, ensuring that the paper’s findings are relevant to the problem statement.

**Opportunities For Improvement:**

1. the baseline tracker is highly overlapped with the tracker proposed in EventVOT (CVPR 2024)
     Event Stream-based Visual Object Tracking: A High-Resolution Benchmark Dataset and A Novel Baseline, CVPR-2024

2. This paper proposes tracking using language, however, the baseline tracker is not developed based on this setting. Thus, I think it is not suitable for this paper.

**Relation To Prior Work:**

Yes, as mentioned in  Figure1: The proposed WebUOT-1M is much larger than existing UOTbenchmarks[37,52,2,1].

**Summary And Contributions:**

The paper introduces WebUOT-1M, the largest public underwater object tracking (UOT) benchmark to date. This dataset is sourced from complex and realistic underwater environments and comprises 1.1 million frames across 1,500 video clips, covering 408 target categories. The authors provide high-quality bounding boxes for underwater targets through meticulous manual annotation and verification. Additionally, WebUOT-1M includes language prompts for video sequences, expanding its application areas, such as underwater vision-language tracking.

To address the challenges of transferring knowledge from open-air domain trackers to UOT, the authors propose a novel omni-knowledge distillation framework based on WebUOT-1M. This framework incorporates various strategies to guide the learning of the student Transformer. To the best of their knowledge, this framework is the first to effectively transfer open-air domain knowledge to the UOT model through knowledge distillation.

The authors comprehensively evaluate WebUOT-1M using 30 deep trackers, showcasing its value as a benchmark for UOT research by presenting new challenges and opportunities for future studies. The completed dataset, codes, and tracking results will be made publicly available.

The main contributions of this work are:

Introduction of WebUOT-1M: The authors introduce WebUOT-1M, the first million-scale benchmark dataset featuring diverse underwater video sequences, essential for offering a dedicated platform for the development and evaluation of UOT algorithms.
Proposal of OKTrack: The authors propose a simple yet strong Omni-Knowledge Distillation Tracking approach, termed OKTrack, for UOT. It is the first work to explore knowledge transfer from a teacher Transformer using underwater and enhanced frames to a student Transformer in the UOT area.
Comprehensive Benchmarking: The authors comprehensively benchmark the proposed approach, along with representative tracking algorithms based on CNN, CNN-Transformer, and Transformer on both the newly proposed WebUOT-1M and existing UOT datasets, obtaining valuable insights.

---

> ### Author Rebuttal · Authors · 2024-08-19
>
> # Response to Reviewer Fqcu [1/6]
>
> Dear Reviewer Fqcu,
>
> We sincerely appreciate your time and effort in reviewing our paper. Firstly, we would like to extend our sincere gratitude for your recognition of the **high quality** of the paper. The paper presents ''a **well-organized and thoughtful approach** to constructing a new underwater object tracking (UOT) benchmark. The authors have **clearly demonstrated their expertise** in the field and have provided a **detailed methodology** for the construction of WebUOT-1M''. We are glad that you feel that our paper is **well-written and easy to understand**, with **clear explanations** of the proposed method and benchmark, and **effectively communicates the purpose and contribution** of the work. We are also grateful for your acknowledgment of our **significant original contribution** to the field of UOT, involving our WebUOT-1M dataset and the proposed method, OKTrack. Your support is highly encouraging. We also appreciate your constructive suggestions on the presentation and experiments in our paper.
>
> ## Opportunities For Improvement:
>
> **1.Comment:** The baseline tracker is highly overlapped with the tracker proposed in **EventVOT (CVPR 2024) Event Stream-based Visual Object Tracking: A High-Resolution Benchmark Dataset and A Novel Baseline, CVPR-2024.**
>
> **Response:**  Thanks for your kindly suggestion. ''Inspired by recent **RGB-event** tracking [63], we propose to distill the knowledge from **enhanced and underwater frames** with the supervision of a pre-trained teacher tracker to a student tracker devoted to handling underwater frames. Due to the limited space, more details are provided in **Appendices**.'' (lines 160-162 of the main text)
>
> We stated in **Appendix F.5 Comparison with Previous Works:** ''In this work, **our primary contribution is the introduction of WebUOT-1M, i.e., the largest and most diverse underwater tracking dataset in terms of target categories and underwater scenarios.** Our dataset covers major underwater scenarios, target categories, and underwater instances in existing UOT datasets [24, 35, 2, 1] and open-air object tracking datasets [15, 14, 21]. The annotated tracking attributes include common attributes (e.g., low resolution, fast motion, and illumination variations) as well as those specific to underwater scenes (e.g., underwater visibility, watercolor variations, and camouflage). Based on the established WebUOT-1M dataset, we further propose a simple yet effective omni-knowledge distillation tracking framework, called OKTrack, for the community. The differences between our approach and existing works (e.g., UVOT400 [1], **HDETrack [38]**, UOSTrack [28]) are listed as follows:''
>
> - ''UVOT400 [1] introduced an underwater tracking dataset consisting of 400 video sequences, 275 K frames, and 17 different tracking attributes and target objects spanning 50 different categories. In comparison, our WebUOT-1M includes 1,500 video sequences, 1.1 million frames, 23 attributes, and 408 target categories. UVOT400 is a partially publicly available dataset, where only the annotations for the first frame of the test set are visible. The complete benchmark, source codes, and tracking results of our work, will be made publicly available.''
>
> - **''HDETrack [38] proposed a multi-modal knowledge distillation strategy for event-based tracking based on RGB frames and event streams. Drawing inspiration from this method, we present an omni-knowledge distillation framework for underwater tracking. In comparison to HDETrack, our approach exhibits several notable differences, such as the addition of token-based contrastive distillation loss and a motion-aware target prediction module.''**
>
> - ''UOSTrack [28] presented a hybrid training strategy using both underwater images and open-air sequences to address sample imbalance, alongside employing motion-based post-processing to mitigate the influence of similar targets. We draw inspiration from its motion-based post-processing to address model drift caused by similar distractors, proposing a MATP module. Compared to UOSTrack, we contribute a new million-scale UOT dataset and demonstrate that training on such a dataset significantly enhances tracking performance, which will benefit the entire UOT community. Additionally, we introduce an omni-knowledge distillation framework for UOT.''
>
> [1] Basit Alawode, Fayaz Ali Dharejo, Mehnaz Ummar, Yuhang Guo, Arif Mahmood, Naoufel Werghi, 431
> Fahad Shahbaz Khan, and Sajid Javed. Improving underwater visual tracking with a large scale dataset and image enhancement. arXiv preprint arXiv:2308.15816, 2023.
>
> [28] Yunfeng Li, Bo Wang, Ye Li, Zhuoyan Liu, Wei Huo, Yueming Li, and Jian Cao. Underwater object tracker: Uostrack for marine organism grasping of underwater vehicles. Ocean Engineering, 285:115449, 2023.
>
> **[38] Xiao Wang, Shiao Wang, Chuanming Tang, Lin Zhu, Bo Jiang, Yonghong Tian, and Jin Tang. Event stream-based visual object tracking: A high-resolution benchmark dataset and a novel baseline. CVPR 2024.**

---

> > ### Author Rebuttal · Authors · 2024-08-19
> >
> > # Response to Reviewer Fqcu [2/6]
> >
> > **2.Comment:** This paper proposes tracking using language; however, the baseline tracker is not developed based on this setting. Thus, I think it is not suitable for this paper.
> >
> > **Response:**  Thanks for your kindly comments. To address your concerns and to demonstrate that the proposed **baseline tracker (OKTrack) is suitable for this paper**, we provide the following reasons. **Since you kindly provided us with some constructive suggestions in the review, please refer to Comment 4 for more explanations and experimental results.**
> >
> > - **Following your suggestions, we have revised the proposed OKTrack to enable it to process both visual and language modalities.** Specifically, we utilize BERT-base [16] as the language encoder and employ the modal mixup [29] as the multi-modal fusion technique. The fused features are then fed into the ViT backbone for feature integration and learning. **This demonstrates that OKTrack is a flexible and scalable baseline tracker that is not only suitable for pure vision underwater object tracking (main attention in this paper), but can also be seamlessly extended to underwater vision-language tracking (a new multi-modal tracking task).**
> >
> > - We have explained why our baseline tracker employs only the visual modality from the perspective of the limited energy consumption and low-latency requirements of underwater low-power devices **(please refer to Appendix F. Additional Discussions: F.2 Why Only Use Underwater Frames for Inference?)**. Furthermore, we have discussed the advantages of our purely visual baseline tracker (OKTrack) in comparison to current SOAT visual trackers and vision-language tracking methods **(please see Appendix F. Additional Discussions: F.4 Why is Our Purely Visual Approach OKTrack Superior to the Current SOAT Visual Trackers and Vision-Language Tracking Methods?)**.
> >
> > - In this work, **our main goal is to build a million-scale underwater target tracking dataset** to ''alleviate the limitations of existing datasets in scale, diversity of target categories and scenarios covered, etc'' (line 3 of the main text). Following existing large-scale open-air tracking datasets (e.g., LaSOT and TNL2K), we manually annotated ''language prompts for video sequences, expanding its application areas, e.g., underwater vision-language tracking'' (lines 10-12 of the main text). **Therefore, as a secondary contribution of this work (compared to our million-scale WebUOT-1M dataset), the proposed visual-based baseline tracker (OKTrack) is also valuable and reasonable as a simple and effective baseline for the UOT task.**
> >
> > - Last but not least, we have performed a pioneering exploration of underwater vision-language tracking through carefully annotated language prompts **(see Section 5.4 Evaluation Results: Underwater Vision-Language Tracking)**. We also note your comment that **"The authors have conducted comprehensive experimental analyses on WebUOT-1M, evaluating the impact of diverse text on tracking performance."**  As a submission to the NeurIPS Datasets & Benchmarks Track, although the initial baseline tracker lacks a language module, we **have evaluated existing vision-language trackers to ensure that our paper is comprehensive and complete.** **Furthermore, considering that underwater object tracking is a far from explored area, our baseline tracker is helpful to other researchers in the community, i.e., confirming its rationality and suitability once again in this paper.**
> >
> >
> > [16] Jacob Devlin, Ming-Wei Chang, Kenton Lee, and Kristina Toutanova. Bert: Pre-training of deep bidirectional transformers for language understanding. In Proceedings of NAACL-HLT, pages 4171–4186, 2019.
> >
> > [29] Mingzhe Guo, Zhipeng Zhang, Heng Fan, and Liping Jing. Divert more attention to vision-language tracking. Advances in neural information processing systems, 2022.

---

> > > ### Author Rebuttal · Authors · 2024-08-19
> > >
> > > # Response to Reviewer Fqcu [3/6]
> > > ## Limitations:
> > > **3.Comment:** Overlap with Existing Work: The baseline tracker proposed in the paper shares a high degree of similarity with the tracker proposed in EventVOT (CVPR 2024). To ensure that the authors’ work is distinct and original, they should either provide a more detailed explanation of how their baseline tracker differs from EventVOT or present new contributions that significantly advance the state of the art beyond what is already known. Constructive Suggestion: The authors could provide **a comparative analysis of their baseline tracker (i.e., OKTrack) with EventVOT (i.e., HDETrack)**, highlighting **unique aspects or improvements that their approach brings to the field**. This would help to establish the novelty and significance of their work.
> > >
> > > **Response:**  Sorry for the unclear description. Although we have already compared and analyzed the differences between our proposed baseline tracker and HDETrack in Appendix F.5 Comparison with Previous Works, we stated that **''HDETrack [38] proposed a multi-modal knowledge distillation strategy for event-based tracking based on RGB frames and event streams. Drawing inspiration from this method, we present an omni-knowledge distillation framework for underwater tracking. In comparison to HDETrack, our approach exhibits several notable differences, such as the addition of token-based contrastive distillation loss and a motion-aware target prediction module''**. To allow the reviewers and readers to gain a deeper understanding of our approach, and to highlight the unique aspects or improvements that our approach brings to the field, we further provide the following statements:
> > >
> > > **(1)** For the loss function, inspired by HDETrack [38], we adopt the three loss functions they use, namely, similarity-based distillation (SKD) loss, feature-based distillation (FKD) loss, and response-based distillation (RKD) loss. It should be noted that the feature-based distillation (FKD) loss and the response-based distillation (RKD) loss are two commonly used losses within the knowledge distillation framework [A]. Furthermore, we introduce a novel Token-based Contrastive Distillation (CKD) loss for underwater object tracking (see Section 4.3 Omni-Knowledge Distillation). **'' The CKD is employed to explicitly align underwater and enhanced tokens by CL [51], aiming to mitigate the distribution discrepancies between multi-view modalities (i.e., underwater and enhanced frames)''** (lines  195-197 of the main text). The above four losses constitute our final Omni-Knowledge Distillation Loss. **Experimental results also validate that the proposed token-based contrastive distillation (CKD) loss enhances the tracking performance of UOT (see Section 5.5 Ablation Study). Combined with the four losses, our final OKTrack significantly outperforms previous SOTA trackers.**
> > >
> > > **[38] Xiao Wang, Shiao Wang, Chuanming Tang, Lin Zhu, Bo Jiang, Yonghong Tian, and Jin Tang. Event stream-based visual object tracking: A high-resolution benchmark dataset and a novel baseline. CVPR 2024.**
> > >
> > > [A] Jianping Gou, Baosheng Yu, Stephen J Maybank, and Dacheng Tao. Knowledge distillation: A survey. International Journal of Computer Vision, 2021
> > >
> > > **(2)** Compared to HDETrack [38], another significant difference in the proposed OKTrack is the Motion-Aware Target Prediction module. We stated in Section 4.2 Network Architecture **''The underwater targets (e.g., fish and sharks) are often surrounded by similar distractors, leading to model drift [42]. To tackle this issue, we design a training-free motion-aware target prediction (MATP) (see Fig. 5) based on Kalman filtering [34]. It involves two primary stages: prediction, which estimates the current state using the previous state, and correction, which combines the estimated state with current observations to determine the optimal state''**. In the **Appendix E. Details of Method: E.1 Motion-aware Target Prediction**, we provided a more detailed description as follows:
> > >
> > > ''The Kalman filtering-based motion-aware target prediction (MATP) [23] is adopted to address tracking drift when the tracker incorrectly locates similar objects. For fast deployment, we borrowed the implementation from SORT [4] and UOSTrack [28]. Readers are strongly recommended to refer to [4] and [28] for more details. We summarize the workflow of MATP as follows:
> > >
> > > - **Candidate Set Extraction:** The search region is divided into n × n patches and the top-N patches with the highest similarity scores to the template are extracted as a candidate set Ct.
> > >
> > > - **Trajectory Prediction:** A Kalman filter (kf) is utilized to predict the target’s position in the current frame, generating the estimation box bE.
> > >
> > > - **Location Score Calculation:** The location score between bE and each candidate box in Ct is calculated using a combination of similarity score and IoU between the boxes.
> > >
> > > - **Match Processing:** The tracker first predicts the target location using detection-based post-processing. If the IoU between the predicted box and bE is below a threshold, the tracker employs motion-based match processing. It calculates the location scores between bE and each candidate box in Ct and outputs the candidate box with the highest score as the tracked target.
> > >
> > > In summary, the MATP leverages trajectory prediction and matching to relocate the target hidden in candidate regions when tracking drift occurs. It effectively utilizes motion information and candidate boxes in each frame, providing a new solution to improve tracking performance on similar object challenges for UOT and beyond. Note that MATP does not require training and is used directly during the inference phase. In **Algorithm 1 (please refer to Appendix E. Details of Method)**, we provide the pseudo-codes of the tracking model to conduct inference with MATP.''

---

> > > > ### Author Rebuttal · Authors · 2024-08-19
> > > >
> > > > # Response to Reviewer Fqcu [4/6]
> > > > **(3)** HDEtrack and the proposed OKTrack address distinctly different tasks. The former primarily addresses the problem of fusing RGB and event streams for RGBE tracking. The latter focuses on the vastly under-explored field of underwater object tracking, which requires addressing challenges such as watercolor variations, camouflage, similar distractors, target blurring, and the severe lack of underwater data, presenting distinctly different challenges from those faced by HDEtrack. **Our method addresses the challenges in underwater object tracking and significantly improves its performance, thus making a significant contribution to the field that cannot be ignored.**
> > > >
> > > > **(4)** **Overall, we have clearly demonstrated our motivations and have provided a detailed explanation of the differences between our proposed OKTrack and previous SOTA trackers (including HDETrack), i.e., highlighting the unique aspects, improvements, and novelty of the proposed OKTrack. We argue that the proposed approach can make significant contributions to the UOT field (this has also been acknowledged by other three Reviewers iASE, 9mKy, and 7sq1).** Once again, we list your kindly and encouraging comments to support the novelty and significance of this work:
> > > >
> > > > - **''Originality: The work presents a significant original contribution to the field of UOT. The authors have successfully addressed the limitations of existing UOT benchmarks by proposing a new approach to generate diverse text annotations using LLMs.''**
> > > > This has a small mistake: In fact, our language annotations are manually labeled by professionals and have undergone multiple rounds of correction (please refer to **Section 3.1 Data Collection and Annotation( ''a professional data labeling team conducts multiple rounds of manual annotation and correction'')**  and **Appendix I. Datasheet**  ("(9) Are there any errors, sources of noise, or redundancies in the dataset? If so, please provide a description. A9: Despite our multiple rounds of careful annotation checks, there may still be some inaccuracies due to occlusion or blurring caused by the movement of underwater targets, such as slight shifts of bounding boxes. **Manually annotated language prompts** might not adequately describe the movement and appearance changes of targets over long video sequences. In the future, we plan to use large multi-modal models to further improve the accuracy and scientific quality of the language annotations)".
> > > >
> > > > - **''Significance: The significance of this work cannot be overstated. WebUOT-1M has the potential to drive substantial advancements in UOT research by providing a more nuanced and challenging dataset for algorithm development and evaluation. The proposed method, OKTrack, has the potential to enhance the performance of UOT algorithms by providing a more representative and diverse dataset.''**
> > > >
> > > >
> > > > **4.Comment:** Integration of Language for Tracking: While the paper proposes tracking using language, the baseline tracker is not developed based on this setting. This inconsistency between the proposed method and the baseline tracker could undermine the overall contribution of the paper. To ensure that the paper’s approach is cohesive and its results are meaningful, the authors should either integrate language into the baseline tracker or justify why the baseline tracker is used without language integration. Constructive Suggestion: The authors could either revise the baseline tracker to include language processing or provide a clear explanation of why language processing is not suitable for their baseline tracker. This would help to clarify the relationship between the proposed method and the baseline tracker, ensuring that the paper’s findings are relevant to the problem statement.
> > > >
> > > > **Response:**  Thanks for your constructive suggestions. Following your suggestions, we offer two explanations to address your concerns: **first, why the baseline tracker is used without language integration (Experience Analysis)**, and **second, what is the impact of revising the baseline tracker to include language processing? (Experiments)**
> > > >
> > > > ### I. Why the baseline tracker is used without language integration? **(Experience Analysis)**
> > > >
> > > > **(1) An intuitive reason is that the language integration will increase calculation overhead.** However, low-power underwater devices often face constraints on computational resources. Therefore, we only consider using video frames as input, without language integration. Specifically, during training, we use both underwater frames and enhanced frames, while during testing, we only use underwater frames **(see Section 4. Methodology, Fig.5 of the main text)**.
> > > >
> > > > **(2) We believe that only using underwater frames (without language integration) for inference can further reduce inference latency.** In **Appendix F.2 Why Only Use Underwater Frames for Inference**, we have performed a detailed discussion and conducted an experiment to demonstrate that tracking on enhanced frames can further improve performance. *"For underwater platforms, e.g., unmanned underwater vehicles, the cameras typically deployed do not have image enhancement capabilities. Adding underwater image enhancement would result in additional energy consumption and latency for these low-power devices. Therefore, a reasonable and low-latency solution is to perform object tracking using only the underwater frames. Moreover, this also follows the evaluation of many existing UOT datasets [24, 35, 5, 1, 2]. As shown in Tab. 3, it is not surprising that tracking on enhanced frames can further improve performance. Therefore, developing lightweight and more effective underwater image enhancement algorithms is also a promising direction."*

---

> > > > > ### Author Rebuttal · Authors · 2024-08-19
> > > > >
> > > > > # Response to Reviewer Fqcu [5/6]
> > > > > **(3) Although our initial baseline tracker lacks language integration, we have discussed the advantages of our proposed baseline in comparison to current SOAT visual trackers and vision-language tracking methods** in Appendix F.4 Why is Our Purely Visual Approach OKTrack Superior to the Current SOAT Visual Trackers and Vision-Language Tracking Methods?
> > > > >
> > > > > *"In our experiments (see the main paper and Tab. 4), we were astonished to find that the proposed 1vision-based approach OKTrack surpassed SOTA visual trackers (e.g., OSTrack [40], SeqTrack-B256 [7], and MixFormerV2-B [10]) even vision-language trackers (e.g., CiteTracker-256 [26], All-in-One [43], and UVLTrack [31]). We speculate that this is due to several factors:"*
> > > > >
> > > > > - "**There is a significant gap between underwater and open-air domains.** Thus, directly applying existing open-air trackers (including visual trackers and vision-language trackers) to underwater environments leads to performance degradation. To quickly verify our hypothesis, we retrained the current SOTA visual tracker (OSTrack) and vision-language tracker (CiteTracker-256) using our WebUOT-1M dataset. The results (see the main paper and **Tab. 4**) show that retraining these open-air trackers on our underwater dataset indeed enhances their tracking performance. This is due to retraining reducing the gap between underwater and open domains. However, the proposed OKTrack still outperforms these retrained open-air trackers (both visual and vision-language trackers). This can be attributed to the effectiveness of the proposed omni-knowledge distillation and MAPT."
> > > > >
> > > > > - "**Limited vision-language tracking datasets.** The existing vision-language tracking datasets (including our WebUOT-1M) are still relatively small (in terms of language annotations), making it challenging to train or fine-tune large visual encoders. It is a promising direction to construct larger-scale vision-language tracking datasets and benchmarks."
> > > > >
> > > > > - "**Ambiguous language annotations.** Most language annotations in existing vision-language tracking datasets primarily describe the target’s state in the initial frame. In long videos and some complex situations, these descriptions fail to accurately convey the target’s current appearance changes. Therefore, training and testing models using the existing language annotations may mislead the models, resulting in decreased tracking performance. A possible solution is to use existing multi-modal large models to generate more accurate language descriptions for the current vision-language tracking datasets, even to produce multi-granularity language descriptions (e.g., concise and detailed descriptions [27]) for a single video sequence."
> > > > >
> > > > > Table 4: Tracking using different prompts. We compare SOTA trackers using language prompt, language prompt+bounding box prompt, and bounding box prompt on WebUOT-1M. ∗ denotes retraining on the WebUOT-1M training set.
> > > > >
> > > > > |Method|Pre (\%) |  nPre (\%) | AUC (\%) |  cAUC (\%)  |  mACC (\%)|
> > > > > | :----:| :----: | :----: | :----: | :----: | :----: |
> > > > > |  | | Language prompt | | | |
> > > > > |JointNLT|   22.4 |   32.2  |  31.2  |   29.8  |   31.2 |
> > > > > |UVLTrack| 22.5 |   33.8  |  31.2  |   30.1  |   31.3|
> > > > > | | | Language prompt + bounding box  | | | |
> > > > > |JointNLT| 25.5 |   34.9  |  32.7  |   31.5  |   32.8|
> > > > > |VLT_SCAR|  33.4 |   44.0  |  37.8 |   36.4 |   38.0|
> > > > > |VLT_TT| 41.7 |   52.1  |  48.3  |   47.3 |   48.8|
> > > > > |UVLTrack| 52.5 |   60.0  |  55.8  |  55.0 |   56.6 |
> > > > > |All-in-One| 53.1 |   61.5 |  57.1  |   56.4  |  58.0 |
> > > > > |CiteTracker-256| 49.3 |   58.4 |  54.6 |  53.7 |   55.2 |
> > > > > |CiteTracker-256| 54.2  |   61.6  |  57.7  |  56.9  |   58.5|
> > > > > | | |Bounding box   | | | |
> > > > > |SeqTrack-B256| 50.8  |    58.5  |  54.0 | 53.3 | 54.7 |
> > > > > |MixFormerV2-B |  45.4  |   54.0   |  51.0 | 50.1 | 51.7  |
> > > > > |OSTrack |  52.9  |   61.1   | 56.5 | 55.8 | 57.4  |
> > > > > |OSTrack*| 55.1  |   61.3   |  57.0 | 56.2 | 57.8  |
> > > > > |OKTrack (Ours) | 57.5  | 63.8  |60.0   |  59.3  |   61.0|
> > > > >
> > > > > **(4) Actually, we used the vision-language tracking model (All-in-One) as the teacher model, implicitly introducing language information through the proposed omni-knowledge distillation.** We stated in Section 5.1 Implementation Details: *"We adopt the unified tracking model [75] as the teacher network. The student network [70] is based on a plain ViT-base backbone, stacked by L (i.e., 12) transformer encoder layers"*. **In the subsequent experiments, we will demonstrate that explicitly fine-tuning the baseline tracker to obtain a visual-language-based baseline tracker (i.e., OKTrack++) can further enhance tracking performance.**
> > > > >
> > > > > [70] Botao Ye, Hong Chang, Bingpeng Ma, Shiguang Shan, and Xilin Chen. Joint feature learning and relation modeling for tracking: A one-stream framework. In European Conference on Computer Vision, pages 341–357, 2022.
> > > > >
> > > > > [75] Chunhui Zhang, Xin Sun, Yiqian Yang, Li Liu, Qiong Liu, Xi Zhou, and Yanfeng Wang. All in one: Exploring unified vision-language tracking with multi-modal alignment. In Proceedings of the 31st ACM International Conference on Multimedia, pages 5552–5561, 2023.

---

> > > > > > ### Author Rebuttal · Authors · 2024-08-19
> > > > > >
> > > > > > # Response to Reviewer Fqcu [6/6]
> > > > > > **(5) As a submission to the NeurIPS Datasets & Benchmarks Track, although the initial baseline tracker lacks a language module, we have evaluated existing vision-language trackers (see Section 5.4 Evaluation Results: Underwater Vision-Language Tracking) to ensure that our paper is comprehensive and complete.** Furthermore, considering that underwater object tracking is a far from explored area, our baseline tracker is helpful to other researchers in the community, i.e., confirming its rationality and suitability in this paper.
> > > > > >
> > > > > > In Section 5.4, we have the following statement: *''Previous UOT datasets lack language prompt annotations [37, 52, 2, 1]. In this work, we perform a pioneering exploration of underwater vision-language tracking through carefully annotated language prompts.  From Tab. 4, we make the following observations. 1) The usage of more cues (e.g., language prompt and bounding box) can significantly boost tracking performance. 2) The language prompt-only methods [45, 81] achieve poor results on WebUOT-1M, similar to existing multi-modal open-air tracking datasets [62, 73, 74], indicating that multi-modal tracking is far from being explored. We expect that the proposed WebUOT-1M dataset can inspire the community to develop multi-modal underwater tracking algorithms.''* (lines 273-280 of the main text)
> > > > > >
> > > > > >
> > > > > > ### II. What is the impact of revising the baseline tracker to include language processing? **(Experiments)**
> > > > > >
> > > > > > - **Following your kindly suggestions, we have revised the proposed OKTrack to enable it to process both visual and language modalities.**  This demonstrates that **OKTrack is a flexible and scalable baseline tracker that is not only suitable for pure vision underwater object tracking (main attention in this paper), but can also be seamlessly extended to underwater vision-language (VL) tracking (a new multi-modal tracking task).**
> > > > > >
> > > > > > - Specifically, we utilize the BERT-base [16] as the language encoder and employ the modal mixup [29] as the multi-modal fusion technique. The fused features are then fed into the ViT backbone for feature integration and learning. We trained OKTrack++ in a similar setup to OKTrack, first using the visual modality to train OKTrack++, and then initializing with the pre-trained weights to further train (finetune) with both visual and language modalities to obtain the final OKTrack++. The reason is that our WebUOT-1M training set is still small compared to existing open-air tracking datasets (e.g., TrackingNet, LaSOT, GOT10k, and TNL2K). Simultaneously training a visual ViT and a language encoder (BERT-base) is challenging to converge. Therefore, we adopted a two-step training strategy for our VL-based OKTrack++.
> > > > > >
> > > > > > [16] Jacob Devlin, Ming-Wei Chang, Kenton Lee, and Kristina Toutanova. Bert: Pre-training of deep bidirectional transformers for language understanding. In Proceedings of NAACL-HLT, pages 4171–4186, 2019.
> > > > > >
> > > > > > [29] Mingzhe Guo, Zhipeng Zhang, Heng Fan, and Liping Jing. Divert more attention to vision-language tracking. Advances in neural information processing systems, 2022.
> > > > > >
> > > > > > - **The results of our two baseline trackers (OKTrack and OKTrack++) are demonstrated in Table A. We find that incorporating the visual modality brings a modest improvement in performance (potentially because of inadequate language annotations), but it significantly increases the model parameters and GPU memory usage, and reduces the inference speed. In the future, we plan to explore more effective vision-language (VL) trackers and use a variety of language annotations to further improve the performance of underwater vision-language tracking.**
> > > > > >
> > > > > >
> > > > > > Table A: Comparison of our two different baseline trackers (OKTrack and
> > > > > >  OKTrack++) using vision and vision-language modalities, respectively. AUC
> > > > > >  /Pre/cAUC/nPre/mACC scores are reported on WebUOT-1M.
> > > > > >
> > > > > > | Method | Type | WebUOT-1M |  #Params | FLOPs |FPS|
> > > > > > | :----:| :----: | :----: | :----: | :----: | :----: |
> > > > > > |OKTrack | Visual-based |   60.0/57.5/59.3/63.8/61.0  |   92.1 M  | 21.5 G |  115|
> > > > > > |OKTrack++ | VL-based |   63.4/58.4/62.9/68.5/64.4  |   150.9 M  | 57.9 G |  66 |
> > > > > >
> > > > > > - **We also report the overall performance and attribute-based performance of the visual-based baseline tracker (OKTrack) and the VL-based baseline tracker (OKTrack++) compared to other SOTA methods on WebUOT-1M. For your convenience, we have presented these results in the attached PDF file. We will update our paper accordingly.**
> > > > > >
> > > > > > Thank you again for reviewing our paper. We hope our response addresses your concerns, and we are happy to answer any further questions you may have.
> > > > > >
> > > > > > Sincerely,
> > > > > >
> > > > > > Authors

---

> > > > > > > ### Comment · Reviewer_Fqcu · 2024-08-26
> > > > > > >
> > > > > > > After reading the response from the authors, my concerns are partly addressed well. I understand the core contributions of this paper are to provide a large-scale benchmark dataset for the tracking task underwater setting. However, the baseline method may be a weak part of this paper, in the final version, I still suggest introducing the language annotations for the tracking, otherwise, why annotate this information if the authors think it is not useful for the tracking underwater? Overall, I think this paper can be accepted due to the key benchmark dataset contributions.

---

> > > > > > > > ### Author Rebuttal · Authors · 2024-08-28
> > > > > > > >
> > > > > > > > # Response to Reviewer Fqcu
> > > > > > > >
> > > > > > > > Dear Reviewer Fqcu,
> > > > > > > >
> > > > > > > > We sincerely appreciate your time and effort in reviewing our paper. We are grateful that you think this paper can be accepted due to the key benchmark dataset contributions. Your support is highly encouraging. We also appreciate your constructive suggestions on the experiments in our paper.
> > > > > > > >
> > > > > > > > - **In the final version, we promise to provide baseline trackers with the visual modality (OKTrack) and both language and visual modalities (OKTrack++), as well as to update the experimental results with language annotations.**
> > > > > > > >
> > > > > > > > - **To facilitate future research, the complete dataset, source codes (including OKTrack and OKTrack++), and tracking results will be made publicly available at [Github](https://github.com/983632847/Awesome-Multimodal-Object-Tracking) after peer review.**
> > > > > > > >
> > > > > > > >
> > > > > > > > Thank you again for your time and helpful suggestions. We are currently organizing the codes and dataset and will open-source our work as soon as possible. Stay tuned!
> > > > > > > >
> > > > > > > >
> > > > > > > > Sincerely,
> > > > > > > >
> > > > > > > > Authors

---

### Comment · Area_Chair_dZN8 · 2024-08-21

Dear Reviewers,

The author has responded to the comments and concerns raised by the reviewers. We kindly ask that you review the author’s responses and provide feedback at your earliest convenience.

Bests,
AC

---

### Comment · Area_Chair_dZN8 · 2024-08-30

Dear Reviewers,

The Discussion Period is ending soon. Reviewers who haven’t responded to the author’s rebuttal yet, please do so as soon as possible. Other reviewers may also use the remaining time to further discuss with the author. Thank you for your efforts.

Bests,
AC

---

### Decision · Program_Chairs · 2024-09-26

**Decision:**

Accept (Poster)

**Comment:**

The paper introduces the WebUOT-1M dataset, which significantly advances underwater object tracking (UOT) research by providing a comprehensive, large-scale benchmark. Key strengths include the scale and diversity of the dataset, which contains 1.1 million frames across 1,500 video clips, offering extensive evaluation opportunities. The authors also propose a novel omni-knowledge distillation framework that enhances the effectiveness of UOT trackers. However, there were some concerns about the overlap with existing methods, the lack of language integration in the baseline tracker, and minor ambiguities in the experiments. Despite these points, the paper’s contributions are substantial, and acceptance is recommended.